# Training Verifiably Robust Agents Using Set-Based Reinforcement Learning

**Manuel Wendl**                                           *manuel.wendl@tum.de*
**Lukas Koller**                                             *lukas.koller@tum.de*
**Tobias Ladner**                                           *tobias.ladner@tum.de*
**Matthias Althoff**                                               *althoff@tum.de*
*Technical University of Munich, Germany*

**Reviewed on OpenReview:** *https://openreview.net/forum?id=DhRPPhSjGA*

## Abstract

Reinforcement learning policies parametrized by deep neural networks have achieved strong performance for continuous control, yet even small input perturbations may lead to unpredictable behavior. This sensitivity limits their use in safety-critical domains, where robustness guarantees are required. Our work addresses this gap between state-of-the-art adversarial training methods and formal verification to train verifiably robust agents. Previous works train networks with individual adversarial perturbations, making them only robust against the specific adversarial attacks used. In contrast, our approach propagates entire perturbed input sets, enclosing all possible adversarial attacks within a single network pass. We leverage this to explicitly penalize the size of the output set (minimizing closed-loop uncertainty) and thereby make the actor robust against all possible attacks. This is realized by the use of set-based policy gradients, where each output within the set has a different gradient, thereby balancing the accuracy and robustness of the network. Doing so, we achieve formal verifiability across different verification frameworks for up to 9 times larger input perturbations compared to standard reinforcement learning and improve certified worst-case performance.

## 1 Introduction

In recent years, deep reinforcement learning using neural networks has significantly improved solving complex control tasks (Mnih et al., 2015; OpenAI et al., 2020; Lillicrap et al., 2016). In many control tasks, state-action spaces are continuous, high-dimensional, and influenced by inherent system uncertainties, modeling errors, and sensor noise (Kober et al., 2013). In practice, we often do not have access to or knowledge about the underlying uncertainty distributions, but have an upper bound of the present perturbations. This poses an important challenge for reinforcement learning when parameterizing policies with neural

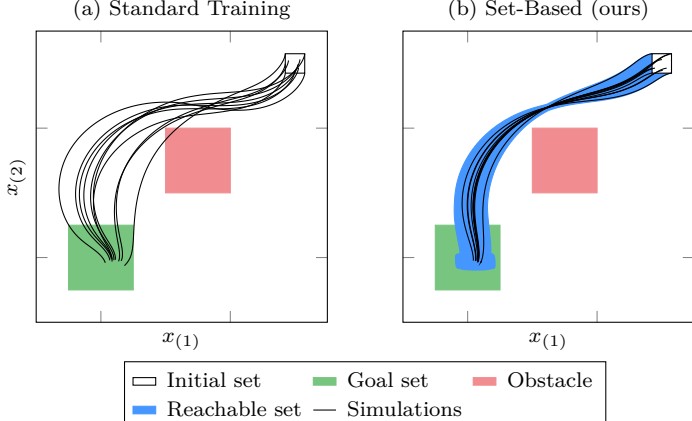

Figure 1: Comparison of standard and set-based training on a navigation task. (a) Trajectories of the standard agent collide with the obstacle. (b) We can formally verify our robust agent.[1]

networks, as they are sensitive to small input perturbations (Szegedy et al., 2014). This may lead to instabilities and safety violations of the controlled system: Fig. 1a shows an example of a navigation

---

[1]Code: *https://cora.in.tum.de/*, Video: *https://youtu.be/gvlLpGr6JK8*

task where small input perturbations lead to trajectories that enter an unsafe set. Robustness to such perturbations is therefore essential. For deployment in safety-critical applications, it is additionally necessary to demonstrate safety and guaranteed worst-case performance. This necessitates formal verifiability under a bounded perturbation budget (Fig. 1b), which is not inherently guaranteed by current state-of-the-art robust reinforcement learning methods. Recently, these formal methods have also been used during training. For example, set-based training (Koller et al., 2025) encloses the set of possible outputs for a given set of inputs at each training step. This enables training a neural network using a gradient set, i.e., each possible output has a distinct gradient. By picking gradients that point toward the center of the output set, the size of the output sets can be controlled. Thereby, the trained neural network is more robust and easier to formally verify as smaller propagated sets reduce the conservatism of the verification algorithm.

In this work, we lift set-based training to reinforcement learning. Our main contributions are:

(i) A novel set-based reinforcement learning algorithm that trains agents that are provably more robust and formally verifiable across different available reachability-analysis frameworks. This is realized by computing a gradient set, which contains a different gradient for each possible output given input perturbations.

(ii) A rigorous analysis of the underlying set propagation to derive a novel set-based loss function for regression tasks, which is used to compute gradient sets that optimize over entire output sets and thus achieve formal verifiability of the trained agents given input perturbations.

(iii) An extensive evaluation including a comparison with state-of-the-art adversarial training algorithms and ablation studies justifying our design choices.

## 2 Preliminaries

### 2.1 Notation

We write vectors as lowercase letters, matrices as uppercase letters, sets as calligraphic letters, and probability distributions as script font letters. The $i$-th entry of $v \in \mathbb{R}^n$ is written as $v_{(i)}$. The entry in the $i$-th row and $j$-th column of a matrix is $M_{(i,j)}$; $M_{(i,\cdot)}$ is the $i$-th row, and $M_{(\cdot,j)}$ the $j$-th column. The horizontal concatenation of matrices $A \in \mathbb{R}^{n \times m}$ and $B \in \mathbb{R}^{n \times p}$ is denoted by $[A\ B]$. We write $|A|$ to denote the absolute values of each entry in $A$, and $\|v\|_p$ denotes the $\ell_p$-norm. The identity matrix is denoted by $I_n \in \mathbb{R}^{n \times n}$, and the vector containing only ones or zeros is denoted by $\mathbf{1}$ or $\mathbf{0}$. The set of natural numbers up to $n \in \mathbb{N}$ is written as $[n] = \{1, 2, \ldots, n\} \subset \mathbb{N}$. We denote a multidimensional interval by $\mathcal{I} = [l, u] = \{x \in \mathbb{R}^n \mid \forall i \in [n]: l_{(i)} \leq x_{(i)} \leq u_{(i)}\}$. Let $\mathcal{S} \subset \mathbb{R}^n$ be a set and $f \colon \mathbb{R}^n \to \mathbb{R}^m$ be a function, then $f(\mathcal{S}) = \{f(x) \mid x \in \mathcal{S}\}$. The gradient of a function $f$ w.r.t. a variable $x$ is denoted by $\nabla_x f(x, \cdot)$. We denote by $\mathrm{diag} \colon \mathbb{R}^n \to \mathbb{R}^{n \times n}$ an operator returning a diagonal matrix with the vector elements on its main diagonal. The expected value of a random variable $x$ under condition $y \sim \mathscr{Y}$ is $\mathbb{E}_{y \sim \mathscr{Y}}[x(y)]$.

### 2.2 Neural Networks

A feed-forward neural network $N_\theta \colon \mathbb{R}^{n_0} \to \mathbb{R}^{n_\kappa}$ with learnable parameters $\theta$ consists of $\kappa \in \mathbb{N}$ alternating linear and activation layers, where the $k$-th layer has $n_k \in \mathbb{N}$ output neurons. The output $\hat{y} = N_\theta(x) \in \mathbb{R}^{n_\kappa}$ is computed by propagating an input $x \in \mathbb{R}^{n_0}$ through all layers.

**Definition 1** (Neural Network, (Bishop & Nasrabadi, 2006, Sec. 5.1)). Given a neural network $N_\theta$ and an input $x \in \mathbb{R}^{n_0}$, the output $\hat{y} = N_\theta(x) \in \mathbb{R}^{n_\kappa}$ is given by

$$
\begin{aligned}
h_0 &= x, \\
h_k &= L_k(h_{k-1}) = \begin{cases} W_k\, h_{k-1} + b_k & \text{if } k\text{-th layer is linear,} \\ \sigma_k(h_{k-1}) & \text{otherwise,} \end{cases} \quad \text{for } k \in [\kappa], \\
\hat{y} &= h_\kappa,
\end{aligned}
$$

with weights $W_k \in \mathbb{R}^{n_k \times n_{k-1}}$, biases $b_k \in \mathbb{R}^{n_k}$, and elementwise activation functions $\sigma_k(\cdot)$.

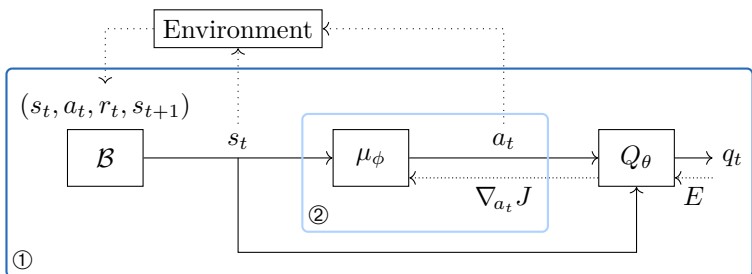

Figure 2: Illustration of the structure of the deep deterministic policy gradient algorithm; ① and ② show the components that are augmented by our set-based training (introduced in Sec. 3).

## 2.3 Deep Deterministic Policy Gradient

We focus on continuous control tasks with a multidimensional state space $\mathcal{S}$ and action space $\mathcal{A}$ (Januszewski et al., 2021; Recht, 2019). Our set-based reinforcement learning approach is based on the deep deterministic policy gradient algorithm (DDPG) (Lillicrap et al., 2016) which consists of an actor $\mu_\phi \colon \mathcal{S} \to \mathcal{A}$ with parameters $\phi$ and a critic $Q_\theta \colon \mathcal{S} \times \mathcal{A} \to \mathbb{R}$ with parameters $\theta$. Starting from an initial state $s_0$, the actor observes the state $s_t \in \mathcal{S}$ at time step $t$ and returns an action $a_t = \mu_\phi(s_t)$, which controls the system until the next time step $t+1$. Using a reward function $r \colon \mathcal{S} \times \mathcal{A} \to \mathbb{R}$ and the state transition probabilities $p(s_{t+1}|s_t, a_t)$, the environment returns a reward $r_t$ and its next state $s_{t+1}$ (Fig. 2). These transitions $(s_t, a_t, r_t, s_{t+1})$ are stored in a buffer $\mathcal{B}$ (Fig. 2). The training objective is to find the policy that maximizes the discounted cumulative reward (Silver et al., 2014, Eq. 8):

$$\max_\phi J(\phi, \rho^\mu) = \max_\phi \mathbb{E}_{s_t \sim \rho^\mu} \left[ \sum_{t=0}^\infty \gamma^t \, r(s_t, \mu_\phi(s_t)) \right], \tag{1}$$

with discount factor $\gamma \in [0, 1]$ and discounted state visitation distribution $\rho^\mu$ for policy $\mu$ (Lillicrap et al., 2016, Sec. 2). The critic $Q_\theta$ approximates the expected total discounted reward for action $a_t$ in state $s_t$ (Lillicrap et al., 2016, Eq. 3):

$$Q_\theta(s_t, a_t) = r(s_t, a_t) + \gamma \, \mathbb{E}_{s_{t+1} \sim \rho^\mu}[Q_\theta(s_{t+1}, \mu_\phi(s_{t+1}))]. \tag{2}$$

The critic is trained off-policy with a stochastic policy $\beta$ (Lillicrap et al., 2016, Eq. 4) and targets $y_t$ (Lillicrap et al., 2016, Eq. 5) using the following bootstrapped Q-iterations:

$$Q_\theta(s_t, a_t) = r(s_t, a_t) + \gamma \, \mathbb{E}_{s_{t+1} \sim \rho^\beta, a_{t+1} \sim \beta}[(Q_\theta(s_{t+1}, \mu_\phi(s_{t+1})))]. \tag{3}$$

The actor is trained using the policy gradient (Lillicrap et al., 2016, Eq. 6):

$$\nabla_\phi J(\phi, \rho^\beta) \approx \mathbb{E}_{s_t \sim \rho^\beta} \left[ \nabla_{a_t} Q_\theta(s_t, a_t) \big|_{a_t = \mu_\phi(s_t)} \nabla_\phi \mu_\phi(s_t) \right]. \tag{4}$$

## 2.4 Set-Based Computing

We model uncertainties using zonotopes due to their favorable computational complexity of the required operations when propagating sets through neural networks.

**Definition 2** (Zonotope (Girard, 2005)). Given a center $c \in \mathbb{R}^n$ and generators $G \in \mathbb{R}^{n \times q}$, we define

$$\mathcal{Z} = \langle c, G \rangle_Z = \{c + G \, \beta \mid \beta \in [-1, 1]^q\} \subset \mathbb{R}^n.$$

The affine map $x \mapsto A \, x + b$ of a zonotope $\mathcal{Z} = \langle c, G \rangle_Z$ is computed by (Althoff, 2010, Sec. 2.4)

$$A \, \mathcal{Z} + b = \{A \, x + b \mid x \in \mathcal{Z}\} = \langle A \, c + b, A \, G \rangle_Z. \tag{5}$$

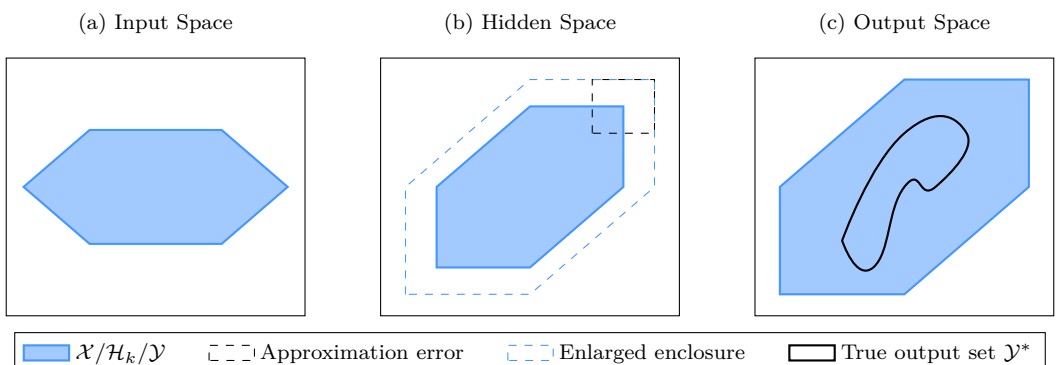

Figure 3: Set-based computing enables us to compute an output set enclosure. Full example in Appendix A.

The enclosing interval of the zonotope $\mathcal{Z} = \langle c, G \rangle_Z$, i.e., $\mathcal{Z} \subseteq [l, u]$ and its diameter $\mathrm{dia}(\mathcal{Z})$, are calculated by (Althoff, 2010, Prop. 2.2)

$$l = c - |G|\mathbf{1}, \quad u = c + |G|\mathbf{1}, \quad \mathrm{dia}(\mathcal{Z}) \coloneqq u - l = 2|G|\mathbf{1}. \tag{6}$$

For some derivations, we write $\mathrm{lnDia}(G) \coloneqq \ln(2|G|\mathbf{1})$ and $\mathrm{lnDia}'(G) \coloneqq \nabla_G \mathrm{lnDia}(G) = \mathrm{diag}(|G|\mathbf{1})^{-1} \operatorname{sign} G$ to avoid clutter, where $\operatorname{sign}(G)$ returns the sign of each entry of matrix $G$. The Minkowski sum of a zonotope $\mathcal{Z} = \langle c, G \rangle_Z$ and an interval $\mathcal{I} = [l, u]$ is computed by (Althoff, 2010, Prop. 2.1 and Sec. 2.4):

$$\begin{aligned}
\mathcal{Z} \oplus \mathcal{I} &= \{x_1 + x_2 \mid x_1 \in \mathcal{Z},\ x_2 \in \mathcal{I}\} \\
&= \langle c + 1/2(u + l), [G \ \mathrm{diag}(1/2(u - l))] \rangle_Z.
\end{aligned} \tag{7}$$

## 2.5 Set-Based Training of Neural Networks

The set-based training of a neural network (Koller et al., 2025) lifts the standard training to a set-based manner, enabling us to (i) propagate the modeled uncertainties $\mathcal{X} \subset \mathbb{R}^{n_0}$ through the neural network and (ii) define a loss function over the computed output set, and (iii) backpropagate the gradient set obtained from the loss function to update the weights of the neural networks. This trains neural networks to be more robust against such input uncertainties.

Computing the exact output set $\mathcal{Y}^* = N_\theta(\mathcal{X}) \subset \mathbb{R}^{n_\kappa}$ of a neural network with ReLU activation functions for $\mathcal{X}$ is NP-hard (Katz et al., 2017). Thus, typically a computationally feasible enclosure of the output set $\mathcal{Y} \supseteq \mathcal{Y}^*$ is computed. This is realized by conservatively propagating the input set through all layers of the neural network using the operations defined in Sec. 2.4, which we also visualize in Fig. 3.

**Proposition 1** (Set-Based Forward Propagation (Singh et al., 2018)). Given an input set $\mathcal{X}$, the output set of a neural network can be enclosed as:

$$\begin{aligned}
\mathcal{H}_0 &= \mathcal{X}, \\
\mathcal{H}_k &= \mathtt{enclose}(L_k, \mathcal{H}_{k-1}), \quad \text{for } k \in [\kappa], \\
\mathcal{Y}^* \subseteq \mathcal{Y} &= \mathcal{H}_\kappa.
\end{aligned}$$

We also write $\mathtt{enclose}(N_\theta, \mathcal{X}) = \mathcal{Y}$ for the output enclosure of the entire network.

Given $\mathcal{Y}$ and a target $t \in \mathbb{R}^{n_\kappa}$, we can define a loss function $E(t, \mathcal{Y})$ over all points in $\mathcal{Y}$ (Koller et al., 2025, Sec. 3.1). Then, $\mathcal{G}_{\mathcal{Y}} = \nabla_{\mathcal{Y}} E(t, \mathcal{Y}) \subset \mathbb{R}^{n_\kappa}$ returns a set of gradients for all points in $\mathcal{Y}$. This gradient set has the same shape as $\mathcal{Y}$, i.e., equal dimensionality and an equal number of generators. Fig. 4 visualizes an example of such a gradient set, where a standard accuracy loss to move the output toward the target is combined with a robustness loss to penalize large – and therefore less robust – output sets. Analogous to Prop. 1, this gradient set can be propagated backward through the network to obtain a gradient set $\mathcal{G}_k$ for each layer $k \in [\kappa]$.

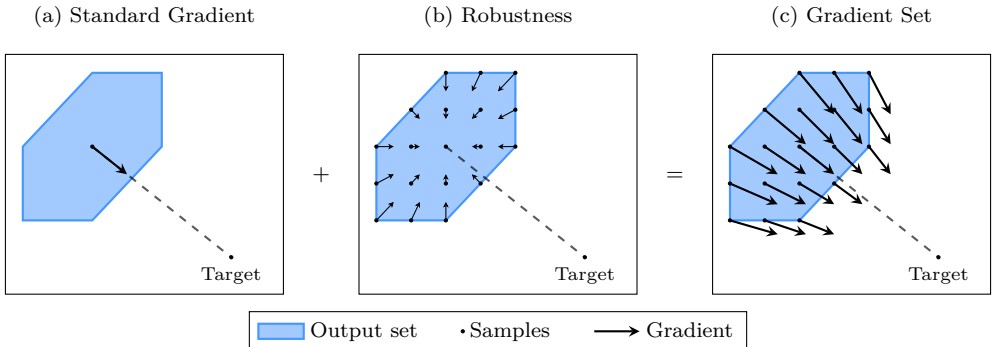

Figure 4: Visualization of a gradient set, where each point in the output set has a different gradient. Here, a penalty on the size of the output set moves each point in the set closer to the center, which, combined with the standard accuracy loss, results in a more robust network.

**Proposition 2** (Set-Based Backward Propagation (Koller et al., 2025, Sec. 4.2)). Given a gradient set $\mathcal{G}_{\mathcal{Y}} \subset \mathbb{R}^{n_\kappa}$ computed with respect to an output set $\mathcal{Y}$ of a neural network, the gradient sets with respect to each layer of the neural network can be enclosed as:

$$\mathcal{G}_\kappa = \mathcal{G}_{\mathcal{Y}},$$
$$\mathcal{G}_{k-1} = \texttt{backpropEnclose}(L_k, \mathcal{G}_k), \quad \text{for } k \in [\kappa],$$
$$\mathcal{G}_{\mathcal{X}} = \mathcal{G}_0.$$

We also write $\texttt{backpropEnclose}(N_\theta, \mathcal{G}_{\mathcal{Y}}) = \mathcal{G}_{\mathcal{X}}$ for the gradient set with respect to the input set $\mathcal{X}$.

These gradient sets are typically computed using information obtained from the forward pass, and can subsequently be used to update the parameters of each layer of the neural network (Koller et al., 2025, Prop. 14). As all operations are computed using basic matrix manipulations, these can be efficiently implemented on a GPU in a batch-wise manner.

## 2.6 Problem Statement

We are given a reinforcement learning task with uncertain initial states $s_0 \in \mathcal{S}_0 \subset \mathbb{R}^n$. Moreover, we model state uncertainties with an $\ell_\infty$-ball of radius $\epsilon \in \mathbb{R}_+$. The set of all perturbations is $\mathcal{V}_\epsilon^\infty := \{\nu \colon \mathbb{R}^n \times \mathbb{R} \to \mathbb{R}^n \mid \forall s_t \in \mathbb{R}^n, \forall t \in \mathbb{R}_+ \colon \|\nu(s_t, t) - s_t\|_\infty \le \epsilon\}$. Our goal is to robustly maximize the reward under adversarial perturbations

$$\max_\phi \min_{\nu \in \mathcal{V}_\epsilon^\infty} J(\phi, \rho^{\mu \circ \nu}), \tag{8}$$

where $\rho^{\mu \circ \nu}$ denotes the state visitation distribution perturbed by adversary $\nu \in \mathcal{V}_\epsilon^\infty$. Furthermore, we provide a guaranteed lower bound of the performance.

# 3 Set-Based Reinforcement Learning

We present a novel approach for training verifiably robust actors using set-based policy gradients. This approach lifts set-based training to reinforcement learning, enabling us to explicitly control the geometry in both size and shape of the policy output set by penalizing undesirable directions and magnitudes. As a result, the conservativeness induced when propagating these sets through the network (Prop. 1) is reduced. This stands in stark contrast to existing robust reinforcement learning approaches, which rely on pointwise perturbations or adversarial samples and thereby never expose the network to the full range of perturbations.

**Set-based actor and critic.** We apply this set-based policy gradient computation to both the actor and the critic (*SA-SC*), which corresponds to a set-based evaluation of ① in Fig. 2. The main steps are shown

---

**Algorithm 1** Set-based reinforcement learning.

---
1: Randomly initialize $Q_\theta$, $\mu_\phi$ with $\theta$, $\phi$
2: Initialize replay buffer $\mathcal{B}$
3: **for** episode $= 1, \ldots,$ maxEpisodes **do**
4:    Get initial observation $s_0$
5:    **for** $t = 0, 1, \ldots,$ maxSteps **do**
6:       `(i) Fill replay buffer` $\mathcal{B}$ `--`
7:       $\mathcal{S}_t \leftarrow \langle s_t, \epsilon I \rangle_Z$                   ▷ Eq. 9
8:       $\mathcal{A}_t \leftarrow \texttt{enclose}(\mu_\phi, \mathcal{S}_t)$                   ▷ Eq. 10
9:       $\tilde{\mathcal{A}}_t \leftarrow \mathcal{A}_t + e_t$, with $e_t \sim \mathbb{P}(\mathcal{E})$                   ▷ Eq. 11
10:       $r_t \leftarrow r(s_t, c_{\tilde{\mathcal{A}}_t})$
11:       $s_{t+1} \leftarrow$ Execute action $c_{\tilde{\mathcal{A}}_t}$
12:       Store transition $(s_t, \tilde{\mathcal{A}}_t, r_t, s_{t+1})$ in $\mathcal{B}$
13:       `(ii) Train networks --`
14:       Sample $n$ transitions from $\mathcal{B}$
15:       Compute targets $y_i$ using $Q_\theta$                   ▷ Eq. 3
16:       Update critic $Q_\theta$                   ▷ Prop. 3
17:       Update actor $\mu_\phi$                   ▷ Def. 4

---

in Alg. 1. As in standard reinforcement learning, Alg. 1 starts by (i) filling up the replay buffer $\mathcal{B}$: Given the current state $s_t$, we enclose all possible adversaries with a zonotope

$$\mathcal{S}_t = \{\nu(s_t, t) \mid \nu \in \mathcal{V}_\epsilon^\infty\} = \langle s_t, \epsilon I \rangle_Z, \tag{9}$$

and enclose the set of possible actions by propagating $\mathcal{S}_t$ through the actor (Prop. 1):

$$\mathcal{A}_t = \langle c_{\mathcal{A}_t}, G_{\mathcal{A}_t} \rangle_Z = \texttt{enclose}(\mu_\phi, \mathcal{S}_t). \tag{10}$$

For the off-policy training of the critic, we store the state and action zonotopes $\mathcal{S}_t$ and $\mathcal{A}_t$ in the replay buffer $\mathcal{B}$ and compute the corresponding set of critic outputs:

$$\mathcal{Q}_t = \langle c_{\mathcal{Q}_t}, G_{\mathcal{Q}_t} \rangle_Z = \texttt{enclose}(Q_\theta, \mathcal{S}_t \times \tilde{\mathcal{A}}_t), \tag{11}$$

where the Cartesian product ($\times$) respects the dependencies between two zonotopes (Lützow & Althoff, 2023, Sec. II.C). The environment receives the center of the action set $c_{\tilde{\mathcal{A}}_t} := c_{\mathcal{A}_t}$ and returns the reward $r(s_t, c_{\tilde{\mathcal{A}}_t})$ and the next state $s_{t+1}$, which are stored in the replay buffer $\mathcal{B}$ as transition $(s_t, \tilde{\mathcal{A}}_t, r_t, s_{t+1})$.

Subsequently, part (ii) starts by training the critic. We randomly sample $n$ transitions from the buffer $\mathcal{B}$. For each transition $i \in [n]$, we compute the target $y_i$ using Eq. 3 and compute a continuous set of gradients for the entire input set using a set-based loss function:

**Proposition 3** (Set-Based Regression Loss). *Given output set $\mathcal{Q}_i = \langle c_{\mathcal{Q}_i}, G_{\mathcal{Q}_i} \rangle_Z \subset \mathbb{R}$ and target $y_i \in \mathbb{R}$, the set-based regression loss is defined as*

$$E_{Reg}(y_i, \mathcal{Q}_i) = \underbrace{1/2(c_{\mathcal{Q}_i} - y_i)^2}_{\text{accuracy loss}} + \underbrace{\eta_Q/\epsilon \ln\mathrm{Dia}(G_{\mathcal{Q}_i})}_{\text{robustness loss}},$$

*with weighting factor $\eta_Q \in \mathbb{R}_+$ and perturbation radius $\epsilon \in \mathbb{R}_+$. The gradient of $E_{Reg}$ w.r.t. $\mathcal{Q}_i$ is:*

$$\nabla_{\mathcal{Q}_i} E_{Reg}(y_i, \mathcal{Q}_i) = \langle c - y_i, \eta_Q/\epsilon \ln\mathrm{Dia}'(G_{\mathcal{Q}_i}) \rangle_Z.$$

*Proof.* The gradient w.r.t. a zonotope is represented as a zonotope as well, consisting of the gradient w.r.t. the center and the gradient w.r.t. the generator matrix (Koller et al., 2025, Def. 8). Hence, the gradient of

the set-based regression is computed by:

$$\begin{aligned}
\nabla_{\mathcal{Q}_i} E_{Reg}(y_i, \mathcal{Q}_i) &= \left\langle \nabla_{c_{\mathcal{Q}_i}} E_{Reg}(y_i, \mathcal{Q}_i), \nabla_{G_{\mathcal{Q}_i}} E_{Reg}(y_i, \mathcal{Q}_i) \right\rangle_Z \\
&= \left\langle c_{\mathcal{Q}_i} - y_i, \frac{\eta_Q}{\epsilon} \operatorname{diag}(|G_{\mathcal{Q}_i}|\,\mathbf{1})^{-1} \operatorname{sign} G_{\mathcal{Q}_i} \right\rangle_Z \\
&= \left\langle c_{\mathcal{Q}_i} - y_i, \frac{\eta_Q}{\epsilon} \operatorname{lnDia'}(G_{\mathcal{Q}_i}) \right\rangle_Z \qquad \qquad \square
\end{aligned}$$

Our set-based regression loss combines an accuracy loss with a robustness loss: The accuracy loss is computed by the half-squared error (Bishop & Nasrabadi, 2006, Eq. 5.14) between the center and the target, thereby improving the accuracy of the network as in regular (point-wise) reinforcement learning. Verifiability is improved by the robustness loss by shrinking the computed enclosure. This is realized by penalizing the diameter, as a computationally tractable surrogate of the volume of the enclosed set (Eq. 6). Together, these losses result in a set of gradients, such that each point in the set is pulled towards the target, as visualized by the gradient set in Fig. 4.

Finally, to train the actor, the combined loss is extended using a set-based policy gradient. As for the critic update, the loss consists of two terms: the accuracy loss, which points to policy improvement, and the robustness loss, which penalizes the diameter of the enclosed set. To update the weights of the actor, we apply Prop. 2 twice, first on the critic network to backpropagate the gradient set to the action space, and subsequently through the actor network to update all weights. Intuitively, this yields similar performance of the actors for perturbed states in the local neighborhood of the zonotope center.

**Definition 3** (Set-Based Policy Gradient ($SA$-$SC$))**.** Given states $\mathcal{S}_i$ with the corresponding actions $\mathcal{A}_i = \langle c_{\mathcal{A}_i}, G_{\mathcal{A}_i} \rangle_Z$ and critic outputs $\mathcal{Q}_i = \langle c_{\mathcal{Q}_i}, G_{\mathcal{Q}_i} \rangle_Z$, a set-based policy gradient is defined as

$$\nabla_{\mathcal{A}_i} J_{Set}(\mu_\phi) := \langle \underbrace{\nabla_{c_{\mathcal{A}_i}} J_{Set}(\mu_\phi)}_{\text{policy gradient}}, \underbrace{\nabla_{G_{\mathcal{A}_i}} J_{Set}(\mu_\phi)}_{\text{robustness}} \rangle_Z,$$

$$\text{where } \nabla_{c_{\mathcal{A}_i}} J_{Set}(\mu_\phi) = \mathbb{E}_{s_i \sim \rho^\beta} \left[ \nabla_{c_{\mathcal{A}_i}} c_{\mathcal{Q}_i} \right],$$

$$\text{and } \nabla_{G_{\mathcal{A}_i}} J_{Set}(\mu_\phi) = -\eta_\mu/\epsilon\, \mathbb{E}_{s_i \sim \rho^\beta} \left[ \omega \operatorname{lnDia'}(G_{\mathcal{A}_i}) + (1 - \omega)\, \nabla_{G_{\mathcal{A}_i}} \operatorname{lnDia'}(G_{\mathcal{Q}_i}) \right],$$

with weights $\eta_\mu \in \mathbb{R}_+$, $\omega \in [0, 1]$ and perturbation $\epsilon \in \mathbb{R}_+$.

In more detail, the set-based policy gradient itself is a zonotope consisting of a center, which corresponds to the standard policy gradient in Eq. 4, and a generator matrix. The generator matrix $\nabla_{G_{\mathcal{A}_i}} J_{Set}(\mu_\phi)$ uses the factor $\omega$ to weight the robustness loss for the action set $\mathcal{A}_i$ and the gradients of the robustness loss of the critic outputs $\mathcal{Q}_i$ in the action space (Fig. 2). Finally, given the gradients w.r.t. the output of the actor and the critic, we can update the respective parameters as described in Sec. 2.5.

**Proposition 4** (Runtime Complexity ($SA$-$SC$))**.** Given an actor $\mu$ and a critic $Q$ with $n_{\max,\mu}$, $n_{\max,Q}$ being the maximum number of neurons per layer, and $n_{\text{total},\mu}$, $n_{\text{total},Q}$ the total number of neurons of the actor and critic, respectively, then the runtime complexity per sample is polynomial and given by

$$\mathcal{O}\big(n_{\max,\mu}^2 n_{\text{total},\mu} \kappa_\mu + n_{\max,Q}^2 (n_{\text{total},\mu} + n_{\text{total},Q}) \kappa_Q\big).$$

As all operations are based on basic matrix manipulations, it can be efficiently implemented on a GPU in a batch-wise manner.

*Proof.* From Sec. 2.5 and the runtime complexity analysis in (Koller et al., 2025, Prop. 17), we observe that the complexity is dominated by the forward and backward propagation of the zonotope, and in particular by the affine maps (Def. 1, Eq. 5). In each nonlinear layer $k$, $n_k$ generators are added to the current zonotope through Eq. 6. Thus, we upper-bound the propagation through the actor $\mu$ with $\mathcal{O}(n_{\max,\mu}^2 n_{\text{total},\mu} \kappa_\mu)$. This upper bound can also be used for the propagation through the critic, with the exception that the input zonotope to the critic already has all generators accumulated by the actor propagation, and thus gives us $\mathcal{O}(n_{\max,Q}^2 (n_{\text{total},\mu} + n_{\text{total},Q}) \kappa_Q)$. $\square$

**Optimizing runtime.** Please note that for $\omega = 1$, the set-based gradient simplifies, and we can omit the set propagation through the critic, since the gradients are solely computed based on the robustness loss of the action set ($SA$-$PC$). In this case, we only require a set-based evaluation of the actor (② of Fig. 2), and the runtime complexity is thus reduced to $\mathcal{O}(n_{\max,\mu}^2 n_{\text{total},\mu} \kappa_\mu)$.

**Definition 4** (Set-Based Policy Gradient ($SA$-$PC$)). Given states $\mathcal{S}_i$ with the corresponding actions $\mathcal{A}_i = \langle c_{\mathcal{A}_i}, G_{\mathcal{A}_i} \rangle_Z$, a set-based policy gradient is defined as

$$\nabla_{\mathcal{A}_i} J_{Set}(\mu_\phi) := \langle \underbrace{\nabla_{c_{\mathcal{A}_i}} J_{Set}(\mu_\phi)}_{\text{policy gradient}}, \underbrace{\nabla_{G_{\mathcal{A}_i}} J_{Set}(\mu_\phi)}_{\text{robustness}} \rangle_Z,$$

$$\text{where } \nabla_{c_{\mathcal{A}_i}} J_{Set}(\mu_\phi) = \mathbb{E}_{s_i \sim \rho^\beta} \left[ \nabla_{c_{\mathcal{A}_i}} Q_\theta(s_i, c_{\mathcal{A}_i}) \right],$$

$$\nabla_{G_{\mathcal{A}_i}} J_{Set}(\mu_\phi) = -\eta_\mu/\epsilon \, \mathbb{E}_{s_i \sim \rho^\beta} \left[ \ln\text{Dia}'(G_{\mathcal{A}_i}) \right],$$

with weight $\eta_\mu \in \mathbb{R}_+$ and perturbation $\epsilon \in \mathbb{R}_+$.

We believe that this novel view of reinforcement learning using set-based computing is a major step toward verifiability, as subsequent verification steps are directly considered in the training process. There is also a connection to the probabilistic view prevalent in reinforcement learning theory, which we discuss in Appendix B.

## 4 Evaluation

We use the MATLAB toolbox CORA (Althoff, 2015; Althoff et al., 2025) to implement our novel set-based reinforcement learning algorithm and compare the $SA$-$PC$ and the $SA$-$SC$ implementation against standard (point-based) training ($PA$-$PC$) and three state-of-the-art adversarial methods: *Naive-*, *Grad-* (Pattanaik et al., 2018, Alg. 2 and 4) and *MAD* (Zhang et al., 2020) (Maximum Action Difference)-based implementations, which compute adversarial attacks to approximate the worst-case observation within a perturbation set. Additionally, we also compare our set-based approach with *RORL* (Yang et al., 2022), an approach based on adversarial smoothing, which has been primarily developed for offline reinforcement learning, but can also be used in the online setting to mitigate out of distribution perturbations. We do not compare *RORL* with our TD3 implementation, since *RORL* is already a method based on a critic ensemble. We consider four diverse benchmarks in our evaluation taken from previous work on robust reinforcement learning (Yuan et al., 2022; Krasowski et al., 2023): *Navigation Task*, *1D Quadrotor*, *Inverted Pendulum*, and *2D Quadrotor*. A detailed description of all benchmarks can be found in Appendix C. Following these works, we use neural networks with ReLU activations and two hidden layers of 64 and 32 neurons for the actor and critic networks. The output layer of the actor has a tanh activation. We provide the mean results and the 95% confidence interval across five different random seeds. Hyperparameters and evaluation details are provided in Appendix C.

**Verified performance.** Following our problem statement (Sec. 2.6), we evaluate our agents based on their worst-case cumulative reward given uncertainties, which we compute using reachability analysis in CORA. Starting at an initial point $s_0$, reachability analysis encloses all reachable states within a time interval $[0, t_{\text{end}}]$ with time horizon $t_{\text{end}} \in \mathbb{R}_+$ (Fig. 1b). The system is subject to uncertainties, which are added at each time step $t \in \{0, 1, \ldots, t_{\text{end}}\}$ (Eq. 9) and carried through until the time horizon is reached. Please note that, in general, this process is outer-approximative to tractably capture all possible trajectories in continuous time and in the presence of nonlinearities within the system and networks. Since also the worst performing solution is contained in this outer-approximation, reachability analysis obtains a formally verified lower bound of the worst-case cumulative reward, which we refer to as *verified performance* from now on. In particular, for reward functions of the form $r(s_t, a_t) = -w^\top |s_t - s^*|$, we can use set-based computing to obtain the verified performance $\underline{V}_\mu(s_0)$ computed from the set of states $\mathcal{S}_t = \langle c_t, G_t \rangle_Z$ obtained by CORA:

$$\underline{V}_\mu(s_0) = \sum_{t=0}^{t_{\text{end}}} -\gamma^t \max_{s_t \in \mathcal{S}_t} w^\top |s_t - s^*| \overset{\text{Eq. 5, Eq. 6}}{=} \sum_{t=0}^{t_{\text{end}}} -\gamma^t \left( w^\top |c_t - s^*| + \text{dia}(w^\top \mathcal{S}_t)/2 \right). \quad (12)$$

Intuitively, $\underline{V}_\mu(s_0)$ is high if our agent is robust against perturbations and drops otherwise.

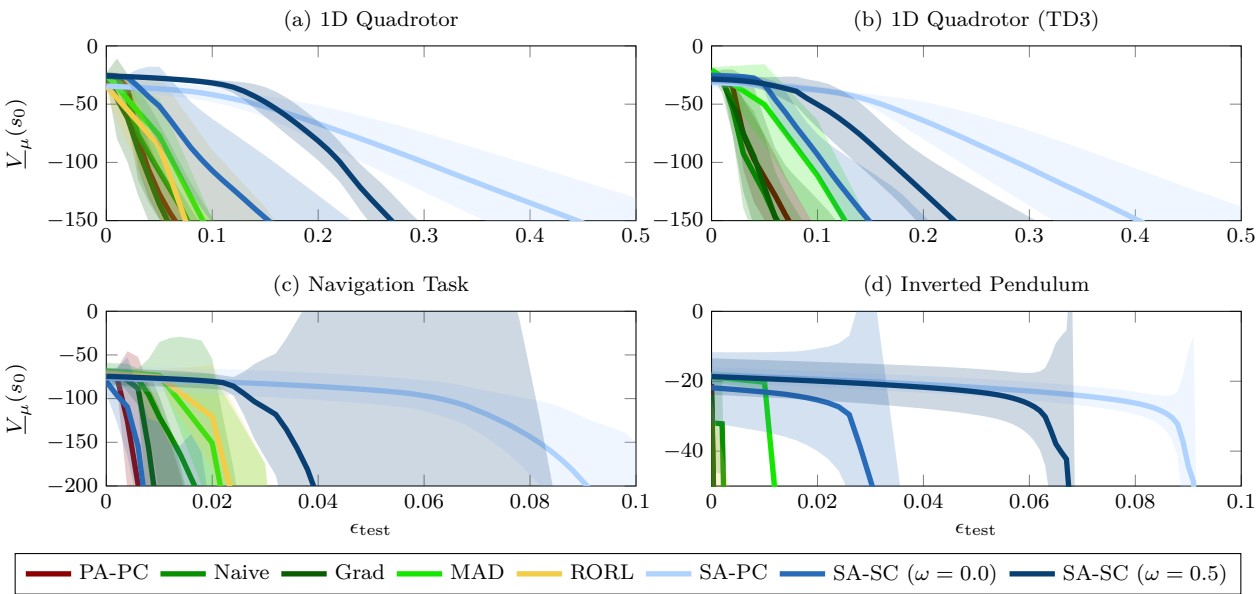

Figure 5: Comparison of verified performance $\underline{V}_\mu(s_0)$ ($\uparrow$) for the (a) *1D Quadrotor*, (c) *Navigation Task*, and (d) *Inverted Pendulum*. The TD3 implementation is compared in (b) for the *1D Quadrotor*.

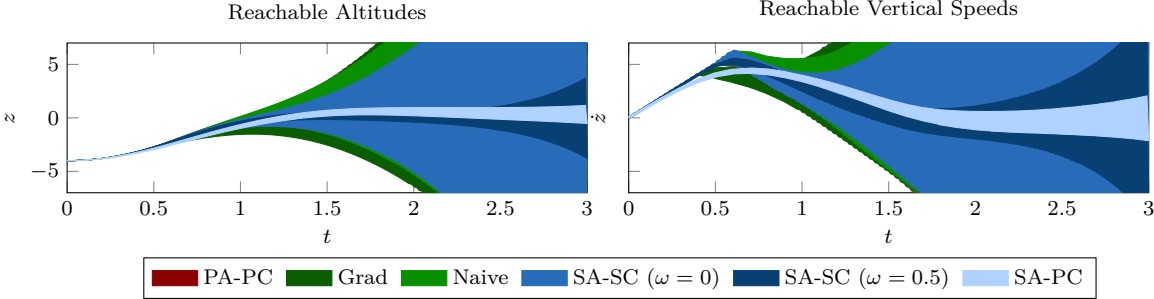

Figure 6: *Quadrotor 1D*: Comparison of reachable altitudes and vertical speeds for $\epsilon_{\text{test}} = 0.15$.

**Main results.** We present the verified performance with increasing perturbation radius $\epsilon_{\text{test}}$ (Eq. 9) of the considered training methods in Fig. 5. The set-based algorithms *SA-PC* and *SA-SC* train agents that can be verified for up to 9 times larger perturbation radii $\epsilon_{\text{test}}$ than the other methods. As $\epsilon_{\text{test}}$ increases, the set-based training methods show a higher verified performance across all benchmarks, indicating reduced performance degradation under growing disturbance (Fig. 5). Notably, the agents are even robust for perturbations above the trained perturbation radius $\epsilon_{\text{train}} = 0.1$ (Tab. 3) on the quadrotor. We showcase the improved robustness in Fig. 6 using a large perturbation radius, where the reachable set of the *SA-PC* agent remains much smaller and thus the stability can be formally verified.

**Ablation study.** Let us continue with our ablation study on various components of our algorithm:

**1) Influence of weighting factor $\omega$ (Def. 3):** Recall that this parameter is used to determine where the diameter of the set is penalized: With $\omega = 0$, only the output set of the critic is penalized (*SA-SC*) and with $\omega = 1$, only the output set of the actor is penalized, which corresponds to the *SA-PC* method. The *SA-PC* implementation trains more conservative actors, which perform best for large $\epsilon_{\text{test}}$ while having a worse verified performance for small $\epsilon_{\text{test}}$. For example, the *SA-PC* actor for the *1D Quadrotor* is the most robust but reaches the goal later using lower vertical velocities (Fig. 6). By fine-tuning $\omega$, a balance can be found where the verified performance for small $\epsilon_{\text{test}}$ can be regained while still being much more robust for larger $\epsilon_{\text{test}}$ (Fig. 5).

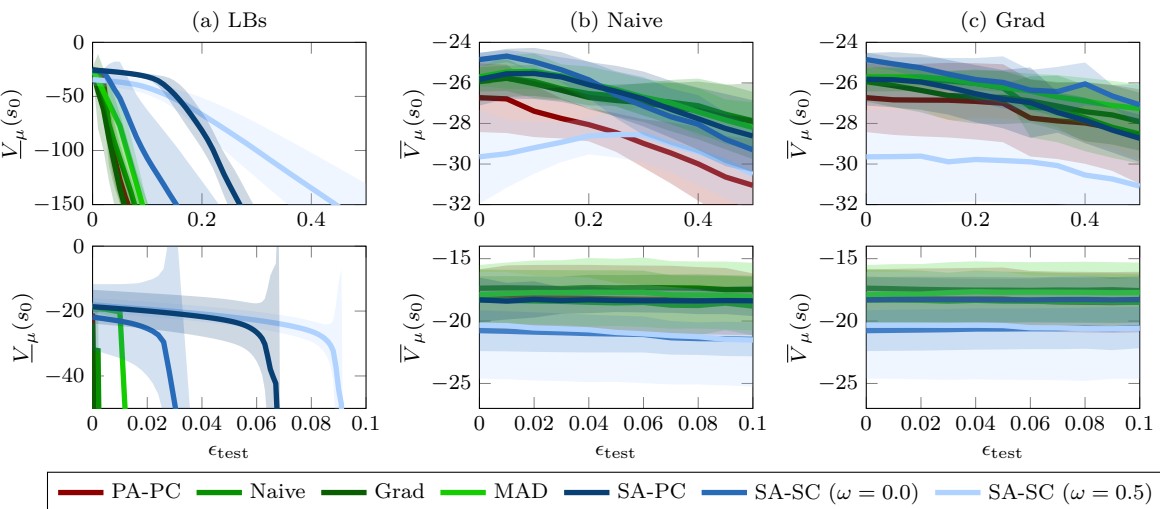

Figure 7: *Quadrotor 1D* (top) and *Inverted Pendulum* (bottom): (a) Comparison of lower bound $\underline{V}_\mu(s_0)$ ($\uparrow$). For comparison, we also include the empirical upper bounds on the worst-case $\overline{V}_\mu(s_0)$ using (b) *Naive* and (c) *Grad* attacks.

Table 1: Time [$s$] for verification of Navigation task agents. Entries with a dash ($-$) indicate that the verification toolbox is not able to verify the system.

| Toolbox | CORA | CROWN-Reach | JuliaReach | NNV |
|---|---|---|---|---|
| Standard | 423.45 | 130.12 | $-$ | $-$ |
| Set-based (ours) | 1.99 | 20.65 | 4.49 | 1903.24 |

**2) Performance under different attacks:** Previous works evaluated their agents solely through adversarial attacks, with learned probabilistic dynamic models (Yang et al., 2024) or randomized smoothing (Wu et al., 2022), which provide an empirical upper bound $\overline{V}_\mu(s_0)$ or a probabilistic bound of the worst-case reward. It is well-known that networks trained using adversarial attacks are robust against their respective attack method but might not be robust against other methods. This ablation study confirms this observation (Fig. 7b-c): Under *Naive* and *Grad* attacks (obtaining $\overline{V}_\mu(s_0)$), their respective trained agents also outperform standard point-based training (*PA-PC*) with DDPG (Lillicrap et al., 2016). However, formally verified performance $\underline{V}_\mu(s_0)$ captures all possible attacks and, thus, *Grad*, *Naive*, and *MAD* methods do not improve on standard point-based training (Fig. 7a) across all benchmarks (Fig. 5). In contrast, agents obtained using our set-based training (*SA-SC* and *SA-PC*) are *formally* robust against the entire set of all possible perturbations (Fig. 7a).

**3) Performance under different verification methods:** Since different formal verification techniques may already be in place for various safety-critical systems, we also show in Tab. 1 that our set-based trained agents can be verified using alternative methods that do not rely on zonotopes. All toolboxes (Althoff, 2015; Xiangru Zhong, 2024; Bogomolov et al., 2019; Tran et al., 2020) verified the set-based agent, whereas only two verified the standard-trained agent, taking much longer to do so. Hence, our set-based agents are easier to verify across different toolboxes due to their robust policy.

**4) Runtime analysis:** Set-based reinforcement learning can be efficiently computed batch-wise on a GPU, but the memory load remains challenging. The different runtimes for 10 learning epochs are listed in Tab. 2 and were run on a server with two *AMD EPYC 7763* 64 core processors, 2 TB RAM, and an *NVIDIA A100-PCIE* 40 GB GPU. For *SA-SC*, the storing of entire action sets with many generators in the replay buffer $\mathcal{B}$ is memory-wise and computationally more expensive. However, *SA-PC* emerges with both competitive training time and high verified performance (Fig. 5).

Table 2: Training time [s/10 epochs] of *SA-PC* and *SA-SC* only show a moderate increase over baselines.

| Benchmark | PA-PC | Naive | Grad | SA-PC | SA-SC |
|-----------|-------|-------|------|-------|-------|
| 1D Quad. | 1.58 | 2.10 | 1.98 | 2.77 | 7.92 |
| Pendulum | 1.82 | 2.04 | 2.02 | 2.87 | 7.66 |
| Nav. Task | 2.35 | 2.89 | 2.84 | 4.35 | 12.43 |

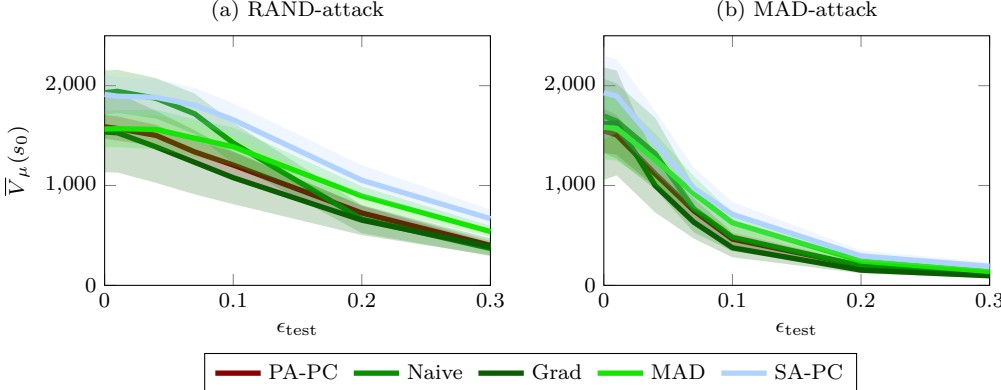

Figure 8: Comparison of the empirical upper bound for the worst-case performance $\overline{V}_\mu(s_0)$ for *Hopper-v2* approximated by (a) uniform random noise and (b) MAD (Zhang et al., 2020).

**5) Extension to ensemble methods:** Our proposed set-based reinforcement learning can be directly extended to the Twin Delayed Deep Deterministic policy gradient algorithm (TD3) (Fujimoto et al., 2018) and other ensemble algorithms (Januszewski et al., 2021). Fig. 5b shows overall a similar performance as with DDPG and for the *SA-PC* version, a better performance for small perturbation radii $\epsilon_{\text{test}}$ of the set-based TD3 algorithm for the *1D Quadrotor*.

**6) Scalability:** In Fig. 8, we show that our algorithm scales to more complex benchmarks with intricate dynamics and larger state and action spaces (Todorov et al., 2012). While the worst-case reward cannot be directly evaluated for every benchmark, we show that for systems that cannot be formally verified by state of the art formal verification toolboxes, our algorithm scales effectively in practice and empirically obtains robust neural networks. We discuss the implementation of the empirical attacks and the benchmark details in Appendix C.3.

**7) Robustness tradeoff:** Our set-based reinforcement learning implementations *SA-PC* and *SA-SC* trade off robustness and performance via the weighting factors $\eta_\mu$ and $\omega$. Increasing $\eta_\mu$ increases robustness, while large $\omega$ values penalize the size of the action set. Additional experiments, hyperparameter ablations, and the entire learning history are provided in Appendix C.

## 5 Related Work

Related works on robust reinforcement learning (Zhang et al., 2021a; Huang et al., 2017; Deshpande et al., 2021; Mandlekar et al., 2017; Lütjens et al., 2020; Zhang et al., 2021b) propose a competitive framework with an adversary (Moos et al., 2022; Pinto et al., 2017): In observation-robust algorithms, the adversary exploits the sensitivity of neural networks to choose a worst-case observation for the policy (Moos et al., 2022). Computing the worst-case observation is often intractable (Madry et al., 2018). Consequently, a variety of naive, gradient-based (Pattanaik et al., 2018; Mandlekar et al., 2017; Huang et al., 2017), learning-based (Zhang et al., 2021a), and convex relaxation methods (Zhang et al., 2020) have been proposed to approximate adversarial observations. For instance, gradient-based methods typically employ the Fast Gradient Sign Method (FGSM) to approximate the most adversarial input (Goodfellow et al., 2015). These adversarial attacks have also been used in conservative smoothing techniques, mainly for offline reinforcement learning, but can also be applied to mitigate out-of-distribution perturbations in an online-learning setting (Yang et al., 2022). In contrast,

our method does not rely on a single adversarial instance. Instead, it updates the agent parameters using the entire set of admissible perturbations, thereby improving robustness and enabling formal verification.

The safe deployment in safety-critical environments requires more than empirical robustness. It demands formal guarantees. While prior work often relies on adversarial attacks or randomized smoothing (Wu et al., 2022) to determine a probabilistic upper bound on the worst-case performance, these approaches do not provide a provable lower bound. In contrast, our work focuses on formal verification of neural network-based control policies. Recent advances (Manzanas Lopez et al., 2023; Kaulen et al., 2025) make it possible to verify entire neural network control systems, and if the obtained reachable set does not violate specifications, the neural network control system is verified as shown in Fig. 1(b).

## 6 Limitations

**Training and verification.** Our approach does not guarantee that the obtained agent is robust in all scenarios, and verification is still required during deployment. Moreover, the formal closed-loop verification of neural network controllers is itself an active research area, and existing toolboxes do not yet support the complex dynamics commonly used in deep reinforcement learning (e.g., MuJoCo benchmarks). For such systems, we can demonstrate empirical robustness improvements through adversarial attacks (Fig. 8), but computing the provable lower bound on the worst-case return is currently out of reach.

**Training runtime and memory.** While the per-sample training cost of our algorithm is polynomial (Prop. 4), it is computationally more expensive compared to standard training due to the repeated zonotope propagations. This effect is most pronounced for *SA-SC*, where the sets need to be propagated through both the actor and the critic. The runtime measurements in Tab. 2 reflect this: *SA-SC* is roughly 3–5 times slower than the point-based baselines, whereas the *SA-PC* variant only has a moderate overhead. We believe such runtime increases are acceptable given the substantially more robust networks we obtain.

**Robustness-performance trade-off.** Our method suffers from the well-known trade-off between accuracy and robustness. For example, *SA-PC* ($\omega = 1$) yields the highest verified performance under large perturbations $\epsilon_{\text{test}}$, but produces more conservative policies that can underperform the point-based baseline for small or zero perturbations – e.g., the *1D Quadrotor* agent reaches the goal more slowly.

**Modeling the system and uncertainties.** The verification assumes white-box access to the system dynamics and bounded uncertainties. While we obtain safety guarantees for the assumed model, we generally cannot show safety if there is a mismatch between the model and the real system. However, recent work on reachset conformance provides methods for identifying models whose reachable sets contain observed real-world measurements (Lützow & Althoff, 2026).

## 7 Conclusion

We introduce the first set-based reinforcement learning algorithm. Unlike other algorithms that rely on adversarial inputs, our approach trains the networks with entire sets of inputs and gradients. As demonstrated by our experiments, this unique advantage makes the resulting networks *verifiably* robust, and does not just improve empirical performance. In contrast, we show that algorithms relying on adversarial inputs only improve against their adversarial attack method. Thus, our trained controllers can be formally verified for large perturbation sets, which is essential for their deployment in safety-critical environments. Consequently, set-based reinforcement learning is an effective, novel approach for training robust neural network controllers. We also show that our set-based learning approach is not limited to the exact setup considered in our work and also generalizes to, e.g., ensemble methods such as TD3. Thus, we believe that our work sparks many research directions to train robust agents that can be deployed in safety-critical systems.

## Acknowledgments

This work was partially supported by the project SPP 2422 (No. 500936349), the project FAI (No. 286525601), and the project SFB 1608 (No. 501798263), all funded by the Deutsche Forschungsgemeinschaft (DFG, German Research Foundation).

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

## A  On Set Propagation Through Neural Networks

We provide additional details on the set propagation from Sec. 2.5 through neural networks. In particular, the operation $\texttt{enclose}(L_k, \mathcal{H}_{k-1})$ (Prop. 1) encloses the output set of the $k$-th layer given the input set $\mathcal{H}_{k-1}$. If the $k$-th layer is linear (Def. 1), the affine map (Eq. 5) is applied:

$$\texttt{enclose}(L_k, \mathcal{H}_{k-1}) = W_k\, \mathcal{H}_{k-1} + b_k. \tag{13}$$

The output set of an activation layer is enclosed by approximating its activation function element-wise using a linear function with slope $m_k \in \mathbb{R}^{n_k}$ and approximation error $[\underline{d}_k, \overline{d}_k]$ (Koller et al., 2025):

$$\texttt{enclose}(L_k, \mathcal{H}_{k-1}) = \text{diag}(m_k)\, \mathcal{H}_{k-1} \oplus [\underline{d}_k, \overline{d}_k], \\ m_k = \frac{\sigma_k(u_{k-1}) - \sigma_k(l_{k-1})}{u_{k-1} - l_{k-1}}. \tag{14}$$

**Example 1.** Given a toy neural network $N_\theta \colon \mathbb{R}^2 \to \mathbb{R}^2$ with $\kappa = 2$ layers, i.e., a linear layer followed by some nonlinear function. We visualize the output of this network for an input set $\mathcal{X}$ in Fig. 3(c):

$$\mathcal{X} = \left\langle \begin{bmatrix} 0 \\ 0 \end{bmatrix}, \begin{bmatrix} 1 & 0.5 & 0.5 \\ 0 & 0.5 & -0.5 \end{bmatrix} \right\rangle_Z. \tag{15}$$

Unfortunately, computing the true output set $N_\theta(\mathcal{X})$ is usually not computationally feasible (Katz et al., 2017), and we thus need to compute an enclosure. In our toy neural network, the linear layer is given by

$$W_1 = \begin{bmatrix} \cos(\phi) & \sin(\phi) \\ -\sin(\phi) & \cos(\phi) \end{bmatrix}, \quad b_1 = \mathbf{0}, \tag{16}$$

where $\phi = -\pi/4$ and thus the layer just rotates $\mathcal{X}$ by 45° counter-clockwise. Please note that this operation does not induce any outer approximation, as

$$\mathcal{H}_1 = W_1 \mathcal{X} + b_1 = \left\langle \begin{bmatrix} 0 \\ 0 \end{bmatrix}, 0.7 \begin{bmatrix} 1 & 1 & 0 \\ 0 & 1 & 1 \end{bmatrix} \right\rangle_Z \tag{17}$$

can be computed exactly using Eq. 5. However, for the nonlinear layer, we need to find an enclosure. To this end, we apply Eq. 14 and choose $m_k = \mathbf{1}$ for simplicity and $[\underline{d}_k, \overline{d}_k] = 0.35[-\mathbf{1}, \mathbf{1}]$. Thus,

$$\mathcal{Y} = \text{diag}(\mathbf{1})\mathcal{H}_1 \oplus 0.7 \left[ \begin{bmatrix} -0.5 \\ -0.5 \end{bmatrix}, \begin{bmatrix} 0.5 \\ 0.5 \end{bmatrix} \right], \tag{18}$$

which results in

$$\mathcal{Y} \subseteq \left\langle \begin{bmatrix} 0 \\ 0 \end{bmatrix}, 0.7 \begin{bmatrix} 1 & 1 & 0 & 0.5 & 0 \\ 0 & 1 & 1 & 0 & 0.5 \end{bmatrix} \right\rangle_Z. \tag{19}$$

Fig. 3c shows that using this enclosure, $\mathcal{Y}^* \subseteq \mathcal{Y}$ holds.

In standard training, these enclosures are not considered, so these post-hoc enclosures can become very conservative. In contrast, set-based training updates the weights considering these enclosures, and if the size of the enclosures is penalized (Fig. 4), not only is the conservativeness of the enclosure reduced, but also the true output set itself – making the networks more robust overall.

## B  Derivation of Set-Based Loss Functions

We now motivate our choices for the set-based loss function (Prop. 3) and our set-based policy gradients (Def. 3 and 4) using a probability perspective. While maximizing the likelihood of outputs is a standard procedure to derive loss functions (Bishop & Nasrabadi, 2006, Sec. 1.2.5), lifting this theory to set-based computing has the unique advantage of integrating formal methods into the training process. To this end, we connect set-based computing and probability theory by assuming a probability distribution over the considered

closed sets. For our derivations, we use a conditional posterior distribution which can be rewritten to be proportional to a likelihood function and a prior distribution (Bishop & Nasrabadi, 2006, Eq. 1.44):

$$\text{cond. posterior} \propto \text{likelihood} \cdot \text{prior.} \tag{20}$$

This reformulation is used as the posterior and is not directly obtainable, but we can assume distributions for the likelihood and the prior to obtain an estimate. In our case, the likelihood corresponds to the standard (point-based) training goal, and the prior penalizes the diameter of the computed sets. The zonotope set representation is point-symmetric with respect to its center. Hence, if we take the most uninformative prior on where a sample is contained in the zonotope, and place a uniform distribution over the generators, we obtain that the expected value over a zonotope is its center:

$$\mathbb{E}_{z \sim \mathcal{Z}}[z] = c. \tag{21}$$

### B.1 Set-Based Regression Loss

We sample random transitions $i \in [n]$ from the buffer $\mathcal{B}$ to obtain a state $s_i$ and the corresponding actions $\tilde{a}_i \sim \tilde{\mathcal{A}}_i$. For each transition, we use Eq. 11 to obtain the critic output $\mathcal{Q}_i$ and use Eq. 3 to obtain the target $y_i$ for each critic output $q_i \sim \mathcal{Q}_i$ using the rewards and next states stored in the buffer. To train $Q_\theta$, we want to maximize the probability $p_\theta(q_i|y_i, s_i, \tilde{a}_i, \beta^{-1})$. Since this probability cannot be computed directly, we model this probability as a conditional posterior using Eq. 20:

$$\underbrace{p_\theta(q_i|y_i, s_i, \tilde{a}_i, \beta^{-1})}_{\text{cond. posterior}} \propto \underbrace{p(y_i|q_i, \beta^{-1})}_{\text{likelihood}} \underbrace{p_\theta(q_i|s_i, \tilde{a}_i)}_{\text{prior}}. \tag{22}$$

For the prior, we assume that $q_i$ is uniformly distributed over the interval $[l_{\mathcal{Q}_i}, u_{\mathcal{Q}_i}] \supseteq \mathcal{Q}_i \subset \mathbb{R}$, as Prop. 1 is also defined over the enclosing interval. Thus, the prior is given by

$$p_\theta(q_i|s_i, \tilde{a}_i) = \mathcal{U}(q_i|l_{\mathcal{Q}_i}, u_{\mathcal{Q}_i}) = \text{dia}(\mathcal{Q}_i)^{-1}. \tag{23}$$

As the critic learns the bootstrapped Q-iterations by regression, we assume that $y_i$ is normally distributed with mean $q_i$ and variance $\beta^{-1}$ with the following likelihood (Bishop & Nasrabadi, 2006, Eq. 1.60):

$$\begin{aligned} p(y_i|q_i, \beta^{-1}) &= \mathcal{N}(y_i|q_i, \beta^{-1}) \\ &= \sqrt{\beta/2\pi} \exp\left(-\beta/2\,(q_i - y_i)^2\right). \end{aligned} \tag{24}$$

Since we observe not a single $q_i$ but an entire set $\mathcal{Q}_i$, we use the expected value $\mathbb{E}_{q_i \sim \mathcal{Q}_i}[q_i] = c_{\mathcal{Q}_i}$ (Eq. 21) in the likelihood function (Bishop & Nasrabadi, 2006, Sec. 10.3). Thus, we obtain the following term to be maximized:

$$p_\theta(q_i|y_i, s_i, \tilde{a}_i, \beta^{-1}) \propto p(y_i|c_{\mathcal{Q}_i}, \beta^{-1})\, p_\theta(q_i|s_i, \tilde{a}_i). \tag{25}$$

Finally, we apply the negative logarithm and set $\beta = (\eta_{\mathcal{Q}}/\epsilon)^{-1}$ to obtain our set-based loss (Prop. 3):

$$\begin{aligned} &- \ln\left(p(y_i|c_{\mathcal{Q}_i}, \beta^{-1})\, p_\theta(q_i|s_i, \tilde{a}_i)\right) \\ &\overset{24,\,23}{=} - \ln\left(\mathcal{N}(y_i|q_i, \beta^{-1})\, \text{dia}(\mathcal{Q}_i)^{-1}\right) \\ &\propto \beta/2\,(c_{\mathcal{Q}_i} - y_i)^2 + \ln\text{Dia}(G_{\mathcal{Q}_i}) \overset{\text{Prop. 3}}{\propto} E_{Reg}(y_i, \mathcal{Q}_i). \end{aligned} \tag{26}$$

We choose $\beta$ that way for easier fine-tuning (Koller et al., 2025, Def. 5).

### B.2 Set-Based Policy Gradient

For a state $s_i$, we derive the set-based policy gradient analogously to (Xiao & Wang, 2022), to maximize the probability of an action $a_i \sim \mathcal{A}_i$ being optimal given $s_i$ – which is again not directly obtainable. Thus, we introduce a binary variable $o_i$ indicating whether $a_i$ is optimal and abbreviate $o_i = 1$ by $o_i$ (Xiao &

Wang, 2022, Sec. 3.1). We again model this probability as a conditional posterior over the action output using Eq. 20:

$$\underbrace{p_{\phi,\theta}(a_i|o_i, s_i, q_i, \alpha)}_{\text{cond. posterior}} \propto \underbrace{p(o_i|q_i, \alpha)}_{\text{likelihood}} \underbrace{p_{\phi,\theta}(a_i|q_i, s_i)}_{\text{prior}}. \tag{27}$$

We assume the likelihood to be exponentially distributed with $\alpha \in \mathbb{R}_+$ (Xiao & Wang, 2022, Sec. 3.2):

$$p(o_i|a_i, q_i, \alpha) = \exp(\alpha^{-1} q_i). \tag{28}$$

The prior in Eq. 27 is not directly computable. Thus, we model it again as a conditional posterior Eq. 20:

$$\underbrace{p_{\phi,\theta}(a_i|q_i, s_i)}_{\text{cond. posterior}} = \underbrace{p_\theta(q_i|a_i, s_i)}_{\text{likelihood}} \underbrace{p_\phi(a_i|s_i)}_{\text{prior}}, \tag{29}$$

where the likelihood function of $q_i$ and the prior for $a_i$ are uniform distributions over the enclosing intervals $[l_{\mathcal{Q}_i}, u_{\mathcal{Q}_i}] \supseteq \mathcal{Q}_i \subset \mathbb{R}$ and $[l_{\mathcal{A}_i}, u_{\mathcal{A}_i}] \supseteq \mathcal{A}_i \subset \mathbb{R}^{n_{\mathcal{A}_i}}$ analogous to Eq. 23:

$$p_\theta(q_i|a_i, s_i) = \mathscr{U}(q_i|l_{\mathcal{Q}_i}, u_{\mathcal{Q}_i}) = \mathrm{dia}(\mathcal{Q}_i)^{-1},$$
$$p_\phi(a_i|s_i) = \mathscr{U}(a_i|l_{\mathcal{A}_i}, u_{\mathcal{A}_i}) = \prod_{j=1}^{n_{\mathcal{A}_i}} \mathrm{dia}(\mathcal{A}_i)_{(j)}^{-1}. \tag{30}$$

Please note that these two probabilities correspond to the evaluation of the actor and the critic, respectively. For the likelihood function of Eq. 27, the expected value $\mathbb{E}_{q_i \sim \mathcal{Q}_i}[q_i] = c_{\mathcal{Q}_i}$ (Eq. 21) is used, and applying the logarithm yields:

$$\ln(p(o_i|c_{\mathcal{Q}_i}, \alpha) \, p_\phi(a_i|s_i) \, p_\theta(q_i|a_i, s_i))$$
$$\overset{28, 30}{=} \alpha^{-1} c_{\mathcal{Q}_i} - \mathbf{1}^\top \ln\mathrm{Dia}(G_{\mathcal{A}_i}) - \ln\mathrm{Dia}(G_{\mathcal{Q}_i}). \tag{31}$$

The set-based policy gradient for SA-SC (Def. 3) is derived by differentiation, where we again set the weighting factor $\alpha = (\eta_\mu/\epsilon)^{-1}$ for easier fine-tuning (Koller et al., 2025, Def. 5). Moreover, we introduce a factor $\omega \in [0, 1]$ to weigh the gradients of the prior terms of $\mathcal{A}_i$ and $\mathcal{Q}_i$.

**Derivation of SA-PC.** For *SA-PC*, only the actor is trained using set-based training while the critic uses standard (point-based) training (① in Fig. 2). Thus, we drop the prior for the critic output $q_i$ as it is no longer evaluated in a set-based fashion and use the expected value $\mathbb{E}_{a_i \sim \mathcal{A}_i}[a_i] = c_{\mathcal{A}_i}$ for the likelihood:

$$\underbrace{p(a_i|o_i, s_i, \alpha, \phi)}_{\text{cond. posterior}} \propto \underbrace{p(o_i|s_i, c_{\mathcal{A}_i}, \alpha)}_{\text{likelihood}} \underbrace{p_\phi(a_i|s_i)}_{\text{prior}}. \tag{32}$$

Taking the logarithm while keeping our assumption on the likelihood function and the prior leads to

$$\ln(p(o_i|s_i, c_{\mathcal{A}_i}, \alpha) \, p_\phi(a_i|s_i))$$
$$\overset{28, 30}{=} \alpha^{-1} Q_\theta(s_i, c_{\mathcal{A}_i}) - \mathbf{1}^\top \ln\mathrm{Dia}(G_{\mathcal{A}_i}). \tag{33}$$

The set-based policy gradient for SA-PC (Def. 4) is derived by differentiation and choosing $\alpha$ as above. This also corresponds to setting $\omega = 1$ in Def. 3.

### B.3 Expectation-Preserving Image Enclosure

In Fig. 9, we plot the probability distributions of a set propagation and visualize the expected value for the first neuron of each layer. We compare the empirically evaluated density of a uniform input disturbance (blue) with the corresponding set-based zonotope propagation (green) across the network layers. The set propagation can be interpreted as a form of variational inference in which the output is constrained to be a zonotope. To illustrate the mean-preserving property of this single-layer variational approximation,

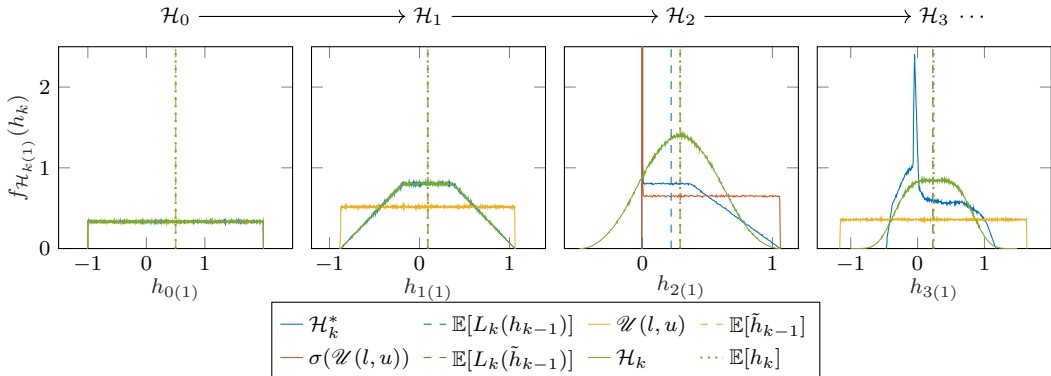

Figure 9: Propagated probability density function $f_{\mathcal{H}_{k(1)}}(h_k)$ of zonotopes for a ReLUneural network: True sampled density function (blue), interval enclosure (yellow), and density of sets from Prop. 1 with $\beta_j \sim \mathcal{U}(-1,1)$ (Def. 2) (green).

we additionally plot the interval enclosures before activation (yellow) and the transformed intervals after activation (red). The expected value is trivially preserved for linear layers as the affine transformation for zonotopes is computed in closed form. For nonlinear layers, we observe that the expected value of the interval enclosure (red vertical line in third plot) is preserved through the enclosure (green vertical line; Prop. 1), with only small deviations from the expected value of the empirically evaluated density function (blue). Let us formally state this observation:

**Proposition 5** (Tight Expectation-Preserving Set Propagation). Given a neural network with ReLU activations and an input set $\mathcal{H}_{k-1}$ with the enclosing interval $[l_{k-1}, u_{k-1}] \supseteq \mathcal{H}_{k-1}$, the expected value of the enclosure of the $k$-th layer is

$$\mathbb{E}_{h_k \sim \mathcal{H}_k}[h_k] = \mathbb{E}_{h_{k-1} \sim \mathcal{U}(l_{k-1}, u_{k-1})}[L_k(h_{k-1})],$$

with $\mathcal{H}_k = \texttt{enclose}(L_k, \mathcal{H}_{k-1})$ having minimal approximation errors.

*Proof.* We split the proof into cases depending on the type of layer $L_k$.

*Case 1 (Linear layer):* The expected value is preserved by linearity of the expectation:

$$\mathbb{E}_{h_k \sim \mathcal{H}_k}[h_k] \overset{21}{=} c_k = W_k\, c_{k-1} + b_k = W_k\, \mathbb{E}_{h_{k-1} \sim \mathcal{U}(l_{k-1}, u_{k-1})}[h_{k-1}] + b_k$$
$$= \mathbb{E}_{h_{k-1} \sim \mathcal{U}(l_{k-1}, u_{k-1})}[W_k\, h_{k-1} + b_k] = \mathbb{E}_{h_{k-1} \sim \mathcal{U}(l_{k-1}, u_{k-1})}[L_k(h_{k-1})].$$

*Case 2 (ReLU layer):* Activation functions are applied element-wise, thus we consider each dimension individually; to avoid clutter, we drop the dimension index $x_{(i)}$. We distinguish between three cases: (2a) $l_{k-1}, u_{k-1} \le 0$, (2b) $l_{k-1}, u_{k-1} \ge 0$, (2c) $l_{k-1} < 0 < u_{k-1}$. For cases (2a) and (2b), ReLU is linear, thus by linearity of the expectation, the expected value is preserved. For case (2c), we approximate the ReLU activation function with the affine map $m_k\, x + t_k$ (Appendix A). The expected value is preserved if the offset $t_k$ satisfies the following condition:

$$
\begin{aligned}
\mathbb{E}_{h_k \sim \mathcal{H}_k}[h_k] = \mathbb{E}[\mathrm{ReLU}(h_{k-1})] \iff & & c_k &= \mathbb{E}[\mathrm{ReLU}(h_{k-1})] \\
\iff & & m_k\, c_{k-1} + t_k &= \mathbb{E}[\mathrm{ReLU}(h_{k-1})] \\
\iff & & t_k &= \mathbb{E}[\mathrm{ReLU}(h_{k-1})] - m_k\, c_{k-1},
\end{aligned}
\tag{34}
$$

with $h_{k-1} \sim \mathcal{U}(l_{k-1}, u_{k-1})$. Hence, we fix the offset $t_k$ w.r.t. the slope $m_k$. To find the optimal slope $m_k$, we compute the expected value $\mathbb{E}[\mathrm{ReLU}(h_{k-1})]$ using the probability density function $f_{\mathcal{H}_k}$ for the distribution of $\mathrm{ReLU}(\mathcal{U}(l_{k-1}, u_{k-1}))$. Therefore, we first compute the probability mass below 0 using the cumulative

distribution function (Hinton & Ghahramani, 1997; Socci et al., 1997):

$$F_{\mathscr{U}(l_{k-1}, u_{k-1})}(0) = \int_{-\infty}^{0} f_{\mathscr{U}(l_{k-1}, u_{k-1})}(h_{k-1}) \, \mathrm{d}h_{k-1}$$

$$= \int_{l_{k-1}}^{0} \frac{1}{u_{k-1} - l_{k-1}} \, \mathrm{d}h_{k-1} = \frac{-l_{k-1}}{u_{k-1} - l_{k-1}}.$$

The probability mass is concentrated in a peak at zero using the Dirac delta $\delta(x)$ (Au & Tam, 1999). Hence, we obtain for a compactly supported function $h(x)$, with respect to the measure $\delta$ the Lebesgue integral:

$$\delta(x) = \begin{cases} \infty & x = 0, \\ 0 & \text{else}, \end{cases} \qquad\qquad \int_{-\infty}^{\infty} h(x)\,\delta(x) \, \mathrm{d}x = h(0).$$

Thus, the probability density function for the post-activation is

$$f_{\mathcal{H}_k}(h_k) = \begin{cases} \frac{1 - l_{k-1}\,\delta(h_k)}{u_{k-1} - l_{k-1}}, & 0 \leq h_k < u_{k-1}, \\ 0 & \text{otherwise}. \end{cases}$$

The resulting density is composed of the uniform distribution for the support $h_k > 0$ and the aggregated probability mass for the negative support $h_{k-1}$ in the Dirac spike at $h_k = 0$. This follows immediately from the definition of $\mathrm{ReLU}(h_{k-1})$. Thus, the expected value of the transformed distribution is given by:

$$\mathbb{E}_{h_{k-1} \sim \mathscr{U}(l_{k-1}, u_{k-1})}[\mathrm{ReLU}(h_{k-1})] = \int_{-\infty}^{\infty} \mathrm{ReLU}(h_{k-1})\, f_{\mathcal{H}_k}(\mathrm{ReLU}(h_{k-1})) \, \mathrm{d}h_{k-1}$$

$$= \int_{0}^{u_{k-1}} h_k\, f_{\mathcal{H}_k}(h_k) \, \mathrm{d}h_k$$

$$= \int_{0}^{u_{k-1}} h_k\, \frac{1 - l_{k-1}\,\delta(h_k)}{u_{k-1} - l_{k-1}} \, \mathrm{d}h_k$$

$$= \frac{h_k^2}{2\,(u_{k-1} - l_{k-1})} \bigg|_{0}^{u_{k-1}} = \frac{u_{k-1}^2}{2\,(u_{k-1} - l_{k-1})}.$$

Plugging this result into Eq. 34 obtains us:

$$t_k = \frac{u_{k-1}^2}{2\,(u_{k-1} - l_{k-1})} - m_k\, c_{k-1} \stackrel{6}{=} \frac{1}{2}\left( \frac{u_{k-1}^2}{(u_{k-1} - l_{k-1})} - m_k\,(u_{k-1} + l_{k-1}) \right). \tag{35}$$

We now find the slope $m_k$ that minimizes the approximation errors, and by additionally satisfying Eq. 34, we ensure preserving the expected value. For concise notation, we abbreviate the approximation error at $x$ with slope $m_k$ by

$$d_x(m_k) := |(m_k\, x + t_k) - \mathrm{ReLU}(x)|. \tag{36}$$

Per dimension, we optimize the slope $m_k$ for minimal approximation errors:

$$\min_{m_k}\ \max_{x \in [l_{k-1}, u_{k-1}]} d_x(m_k).$$

Using (Koller et al., 2025, Prop. 7), we know that the approximation error is located at either $x \in \{l_{k-1}, 0, u_{k-1}\}$ and rewrite:

$$\min_{m_k}\ \max_{x \in \{l_{k-1}, 0, u_{k-1}\}} d_x(m_k).$$

From $l_{k-1} < 0 < u_{k-1}$, we can simplify

$$d_{l_{k-1}}(m_k) = |m_k\, l_{k-1} + t_k|,$$
$$d_0(m_k) = |t_k|, \tag{37}$$
$$d_{u_{k-1}}(m_k) = |m_k\, u_{k-1} + t_k - u_{k-1}|.$$

Moreover, we know that at least one approximation error is greater than 0:

$$d_{l_{k-1}}(m_k) > 0 \vee d_0(m_k) > 0 \vee d_{u_{k-1}}(m_k) > 0.$$

Hence, the optimal slope $m_k$ is located at an intersection of the error functions $d_{l_{k-1}}(m_k)$, $d_0(m_k)$, $d_{u_{k-1}}(m_k)$. Thus, for each intersection we find the optimal slope:

*Case (2c.i):* $d_{l_{k-1}}(m_k) = d_0(m_k)$

$$d_0(m_k) = d_{l_{k-1}}(m_k)$$

$$\overset{37}{\Longleftrightarrow} \qquad |t_k| = |m_k\, l_{k-1} + t_k|$$

$$\Longleftrightarrow \qquad t_k^2 = (m_k\, l_{k-1} + t_k)^2$$

$$\Longleftrightarrow \qquad 0 = (m_k\, l_{k-1})^2 + 2\, m_k\, l_{k-1}\, t_k$$

$$\overset{35}{\Longleftrightarrow} \qquad 0 = m_k^2 - \frac{u_{k-1}}{u_{k-1} - l_{k-1}}\, m_k$$

$$\Longleftrightarrow \qquad m_k = 0 \vee m_k = \frac{u_{k-1}}{u_{k-1} - l_{k-1}}.$$

*Case (2c.ii):* $d_{l_{k-1}}(m_k) = d_{u_{k-1}}(m_k)$

$$d_{l_{k-1}}(m_k) = d_{u_{k-1}}(m_k)$$

$$\overset{37}{\Longleftrightarrow} \qquad |m_k\, l_{k-1} + t_k| = |m_k\, u_{k-1} + t_k - u_{k-1}|$$

$$\Longleftrightarrow \qquad (m_k\, l_{k-1} + t_k)^2 = (m_k\, u_{k-1} + t_k - u_{k-1})^2$$

$$\overset{35}{\Longleftrightarrow} \qquad m_k\, u_{k-1}\, l_{k-1} = u_{k-1}\, \frac{u_{k-1}^2}{u_{k-1} - l_{k-1}} - u_{k-1}^2$$

$$\Longleftrightarrow \qquad m_k = \frac{u_{k-1}}{u_{k-1} - l_{k-1}}.$$

*Case (2c.iii):* $d_0(m_k) = d_{u_{k-1}}(m_k)$

$$d_0(m_k) = d_{u_{k-1}}(m_k)$$

$$\overset{37}{\Longleftrightarrow} \qquad |t_k| = |m_k\, u_{k-1} + t_k - u_{k-1}|$$

$$\Longleftrightarrow \qquad t_k^2 = (m_k\, u_{k-1} + t_k - u_{k-1})^2$$

$$\overset{35}{\Longleftrightarrow} \qquad -m_k^2\, l_{k-1} + m_k \left( \frac{l_{k-1}\, (2\, u_{k-1} - l_{k-1})}{u_{k-1} - l_{k-1}} \right)$$

$$= \frac{u_{k-1}^2}{u_{k-1} - l_{k-1}} + u_{k-1}$$

$$\Longleftrightarrow \qquad m_k = 1 \vee m_k = \frac{u_{k-1}}{u_{k-1} - l_{k-1}}.$$

In each case, we show that the proposed slope in Eq. 38 is optimal and minimizes the approximation errors $\underline{d}_k, \overline{d}_k \in \{d_{l_{k-1}}, d_0, d_{u_{k-1}}\}$. Please note that for all three cases,

$$m_k = \frac{\text{ReLU}(u_{k-1}) - \text{ReLU}(l_{k-1})}{u_{k-1} - l_{k-1}} = \frac{u_{k-1}}{u_{k-1} - l_{k-1}}. \tag{38}$$

$\square$

Using Prop. 5, we simplify the likelihood in Eq. 25, Eq. 31 and Eq. 33 with the expected value of the neural network output, i.e., the center of the enclosure.

## C  Evaluation Details

### C.1  Benchmark Descriptions

Let us briefly describe the specifications of the used benchmarks in this section.

*1D Quadrotor:* The state is $s = \begin{bmatrix} z & \dot{z} \end{bmatrix}^\top$, with altitude $z$, velocity $\dot{z}$, and dynamics (Yuan et al., 2022):

$$\dot{s} = \begin{bmatrix} \dot{z} & \frac{a+1}{2\,m} - g \end{bmatrix}^\top, \tag{39}$$

with action space $a \in [-1, 1]$, standard gravity $g = 9.81$, and mass $m = 0.05$. Starting from the set of initial states $s_0 \in [\begin{bmatrix} -4 & 0 \end{bmatrix}^\top, \begin{bmatrix} 4 & 0 \end{bmatrix}^\top]$, the Quadrotor is stabilized at $s^* = \mathbf{0}$; the reward function is $r(s_t, a_t) = -\begin{bmatrix} 1 & 0.01 \end{bmatrix}|s_{t+1} - s^*|$ for a time horizon of $3s$.

*Navigation Task:* We use a unicycle model with state $s = \begin{bmatrix} x & y & \theta & v \end{bmatrix}^\top$ and dynamics (Lopez et al., 2022):

$$\dot{s} = \begin{bmatrix} v\cos(\theta) & v\sin(\theta) & a_{(1)} & a_{(2)} \end{bmatrix}^\top, \tag{40}$$

with action space $a \in [-\mathbf{1}, \mathbf{1}]$. From the initial state $s_0 = \begin{bmatrix} 3 & 3 & 0 & 0 \end{bmatrix}^\top$, the task is to navigate to the goal $s^* = \mathbf{0}$ without colliding with an obstacle $\mathcal{O} = [\mathbf{1}, 2 \cdot \mathbf{1}]$. The reward function is $r(s_t, a_t) = -\begin{bmatrix} 1 & 1 & 0 & 0 \end{bmatrix}|s_{t+1} - s^*| - c$, with $c = 1$ if $s_{t+1} \in \mathcal{O}$ and otherwise $c = 0$. We consider a time horizon of $8s$.

*Inverted Pendulum:* The state is $s = \begin{bmatrix} \theta & \dot{\theta} \end{bmatrix}^\top$, with angle $\theta$, angular velocity $\dot{\theta}$, dynamics (Krasowski et al., 2023)

$$\dot{s} = \begin{bmatrix} \dot{\theta} & \frac{g}{l}\sin(\theta) + \frac{1}{m\,l^2}a \end{bmatrix}^\top, \tag{41}$$

action space $a \in [-15, 15]$, standard gravity $g = 9.81$, mass $m = 1$, and length $l = 1$. The goal is to stabilize the pendulum in the upright position $s^* = \mathbf{0}$; the reward function is $r(s_t, a_t) = -\begin{bmatrix} 1 & 0.01 \end{bmatrix}|(s_{t+1} - s^*)|$ for a time horizon of $3s$.

*2D Quadrotor:* The state of the system is defined as $s = \begin{bmatrix} x & \dot{x} & z & \dot{z} & \theta & \dot{\theta} \end{bmatrix}^\top$, with horizontal displacement $x$, horizontal velocity $\dot{x}$, altitude $z$, vertical velocity $\dot{z}$, angle $\theta$, angular velocity $\dot{\theta}$ and dynamics (Yuan et al., 2022):

$$\dot{s} = \begin{bmatrix} \dot{x} \\ \sin(\theta)\frac{\tilde{a}_{(1)}+\tilde{a}_{(2)}}{m} \\ \dot{z} \\ \cos(\theta)\frac{\tilde{a}_{(1)}+\tilde{a}_{(2)}}{m} - g \\ \dot{\theta} \\ \frac{l(\tilde{a}_{(2)}-\tilde{a}_{(1)})}{\sqrt{2}J_y} \end{bmatrix}, \tag{42}$$

where $\tilde{a} = (1 + \frac{1}{2}a)\frac{m\,g}{2}$ with action $a \in [-\mathbf{1}, \mathbf{1}] \subset \mathbb{R}^2$. The constant $g = 9.81$ is the standard gravity, $m = 0.027$ is the mass of the Quadrotor, $l = 0.0397$ and $J_y = 1.4 \cdot 10^{-4}$ respectively define the arm length of the propeller mount and the moment of inertia around the $y$ axis. The reward function is given by $r(s_t, a_t) = -\begin{bmatrix} 1 & 0.01 & 1 & 0.01 & 0 & 0 \end{bmatrix}|s_{t+1} - s^*|$ for a time horizon of $3s$, with the goal to stabilize the Quadrotor at $s^* = \mathbf{0}$.

*MuJoCo Hopper-v2:* This benchmark simulates a one-legged robot hopping forward as efficiently as possible. As providing the full set of equations for this benchmark would be too space consuming here, we refer interested readers to (Todorov et al., 2012).

### C.2  Hyperparameters

We list the hyperparameters used to train the networks in Tab. 3.

Table 3: Training parameters for *PA-PC*, *SA-PC* and *SA-SC*.

| Parameter | DDPG | TD3 |
|---|---|---|
| Actor learning rate | $1 \cdot 10^{-4}$ | $1 \cdot 10^{-4}$ |
| Critic learning rate | $1 \cdot 10^{-3}$ | $1 \cdot 10^{-3}$ |
| Critic $L_2$ weight regularization $\lambda_Q$ | 0.01 | 0 |
| Discount factor $\gamma$ | 0.99 | 0.99 |
| Target update factor $\tau$ | 0.05 | 0.05 |
| Exploration noise std. deviation $\sigma$ | 0.1 | 0.1, 0.2 |
| Batchsize | 64 | 64 |
| Buffersize | $1 \cdot 10^6$ | $1 \cdot 10^6$ |
| Episodes | 2000 | 2000 |
| Perturbation radius $\epsilon_{\text{train}}$ | 0.1 | 0.1 |
| Actor weighting factor $\eta_\mu$ | 0.1 | 0.1 |
| Critic weighting factor $\eta_Q$ | 0.01 | 0.01 |

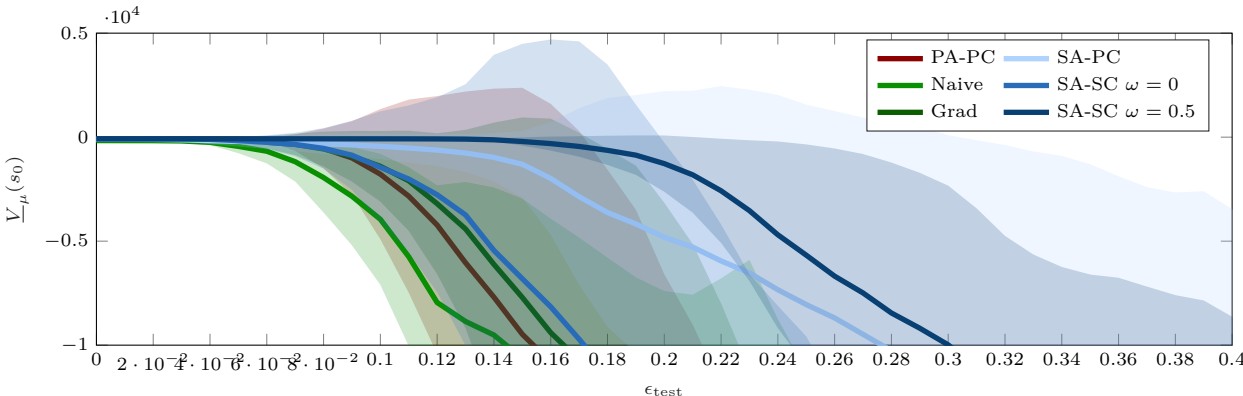

Figure 10: Comparison of $\underline{V}_\mu(s_0)$ for Quadrotor 2D.

## C.3 Additional Experiments

**Quadrotor 2D:** We provide additional experiments on the *2D Quadrotor* and the MuJoCo *Hopper-v2* benchmark (Todorov et al., 2012) and again average the results over the last five agents in each training run and compute the mean and a 95% confidence interval across five independent random seeds. Fig. 10 compares the lower bounds $\underline{V}_\mu(s_0)$ of the different training algorithms.

**Locomotion:** Since, to the best of our knowledge, no existing verification toolbox can compute reachable sets for locomotion benchmarks, we are unable to provide a formal lower bound for *Hopper-v2*. However, we want to stress that this is not a limitation of our training approach but rather of the subsequent verification step. Thus, to demonstrate the scalability of our approach to such benchmarks, we approximate $\underline{V}_\mu(s_0)$ using 50 trajectories under the MAD-attack (Zhang et al., 2020) and 200 trajectories perturbed with uniform random noise sampled from the $\ell_\infty$ ball with radius $\epsilon_{\text{test}}$. The corresponding worst-case returns are shown in Figure Fig. 8 (MAD attack and random noise). For *Hopper-v2*, these empirical estimates provide an upper bound on the worst-case performance and demonstrate the scalability of our method. Notably, even under noise-free conditions, the worst returns of SA-PC-trained agents, evaluated over randomly initialized trajectories, consistently exceeded those of agents trained with a point-wise robustness criterion. We additionally provide videos illustrating agent behaviors under the uniform-random[2] and MAD[3] attacks using the first random seed, with $\epsilon_{\text{test}} = 0.1$ and $\epsilon_{\text{test}} = 0.075$. To better visualize the failure modes of agents, we disable early

---

[2]Video: `https://github.com/ManuelWendl/VerifiablyRobustSetBasedRL/blob/master/assets/videosRand.mp4`

[3]Video: `https://github.com/ManuelWendl/VerifiablyRobustSetBasedRL/blob/master/assets/videosMad.mp4`

Table 4: Training time [s/10 epochs] of *SA-PC* shows a moderate increase over baselines on Hopper-v2.

| Benchmark | PA-PC | Naive | Grad | MAD | SA-PC |
|-----------|-------|-------|------|-----|-------|
| Hopper-v2 | 44.70 | 58.88 | 69.97 | 134.07 | 343.56 |

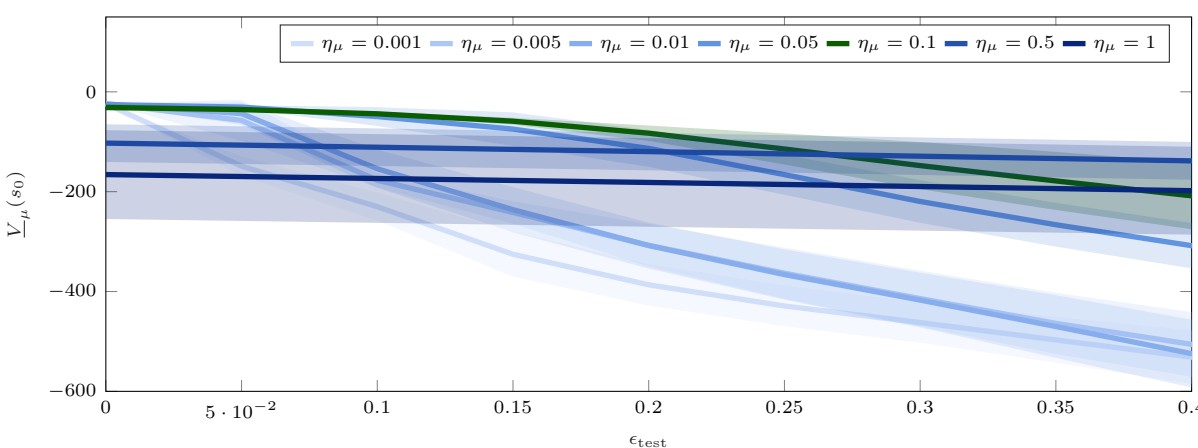

Figure 11: Trade-off in verified performance for hyper parameter ablation $\eta_\mu$.

termination in these demonstrations. The results clearly show that the MAD attack is more effective at degrading agent performance compared to the uniform-random baseline. Notably, across both attack types, our *SA-PC* agent consistently demonstrates superior robustness and overall performance. We additionally report in Tab. 4 the run time required for ten episodes of training the locomotion benchmark Hopper-v2 for the different baselines and *SA-PC*. We report the mean runtimes over the 5 seeds, run on an Intel Core Ultra 9 185H CPU with an NVIDIA RTX 3000 Ada GPU.

**Hyperparameter Discussion:** Set-based reinforcement learning introduces the additional parameters $\eta_\mu$ and $\eta_Q$, which appear in the set-based policy gradient in Def. 3 and the set-based regression loss in Prop. 3. In Tab. 3, we list the hyperparameters used in our benchmarks. Notably, $\eta_\mu$ and $\eta_Q$ are fixed in all experiments. We ease tuning these parameters by choosing $\beta$ and $\alpha$ in Eqs. 26 and 31 as proposed by Koller et al. (2025). For a given perturbation radius $\epsilon_{\text{test}}$, we study the effect of varying $\eta_\mu$ for *SA-PC* on the 1D Quadrotor benchmark. The hyperparameter $\eta_\mu$ directly scales the set-based gradient of the zonotope generators. As shown in Fig. 11, increasing $\eta_\mu$ improves robustness: verified performance near the noise-free case $\epsilon_{\text{test}} = 0$ decreases slightly for larger $\eta_\mu$, whereas verified performance at larger $\epsilon_{\text{test}}$ increases. This matches the intended trade-off between nominal (zero-perturbation) performance and robustness to larger perturbations. We therefore conclude that the choice of $\eta_\mu = 0.1$ (green) performs well, since it still achieves high verified rewards for large $\epsilon_{\text{test}}$, while it preserves good rewards for the noise-free case $\epsilon_{\text{test}} = 0$. We also report the learning history for different hyperparameter settings in Fig. 12. Moderate values of $\eta_\mu$ lead to stable convergence, while very large $\eta_\mu$ slow or prevent convergence to the optimum.

We next conduct an ablation study on the hyperparameter $\eta_Q$ for *SA-SC*, keeping $\eta_\mu = 0.1$ fixed. As shown in Fig. 13, large values of $\eta_Q$ (e.g. $\eta_Q = 1$) make the value function highly contractive, which leads to a reduced performance for small perturbation radii $\epsilon_{\text{test}}$ and to convergence difficulties, as illustrated in Fig. 14. As $\eta_Q$ decreases, convergence improves and the value function captures more of the variability induced by observation noise. Consequently, the actor learns a more conservative policy, resulting in higher verified performance for larger $\epsilon_{\text{test}}$. We find that $\eta_Q = 0.01$ (green) provides the best verified performance across all perturbation radii. Further reducing $\eta_Q$ causes verified performance to decline again.

**Learning History:** In Fig. 16, we present the full learning curves, which show that set-based reinforcement learning exhibits convergence behavior comparable to the adversarial baselines. For clarity, we report the rewards evaluated without observation noise during training. Additionally, we report the learning curves for the Locomotion benchmark (Hopper-v2) from Fig. 8 in Fig. 15. For this analysis, we evaluate both

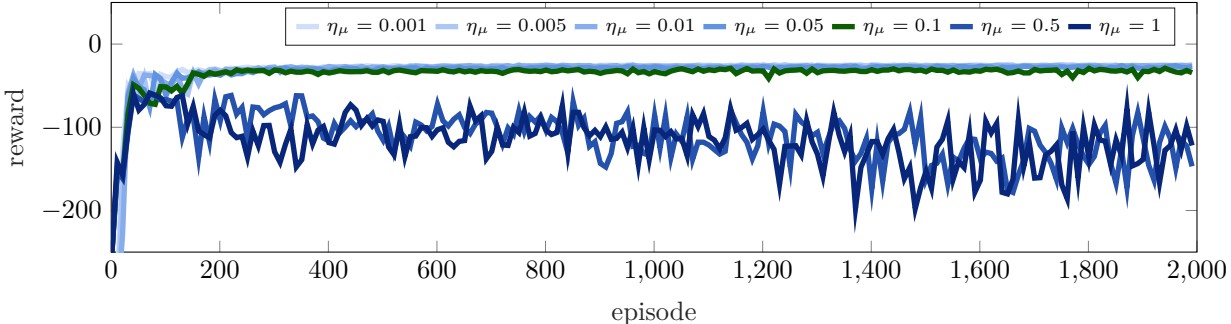

Figure 12: Learning behavior for hyper parameter ablation $\eta_\mu$.

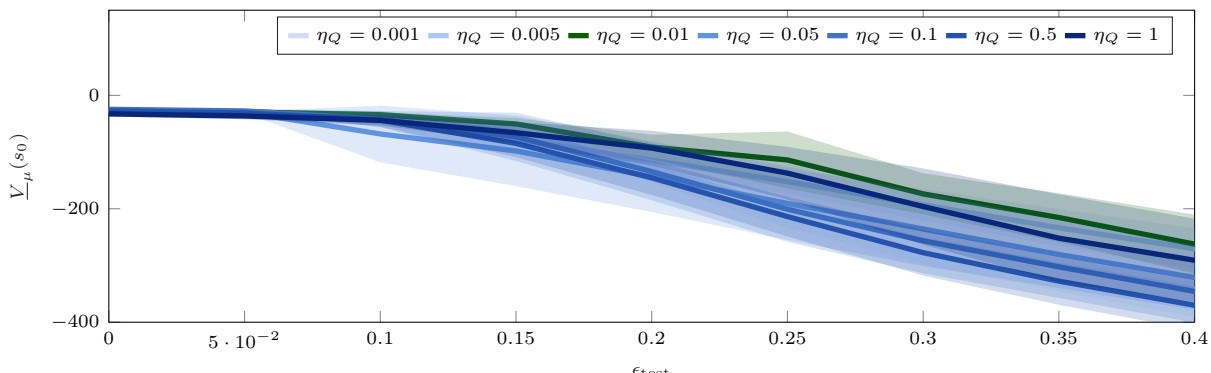

Figure 13: Hyper parameter ablation $\eta_Q$.

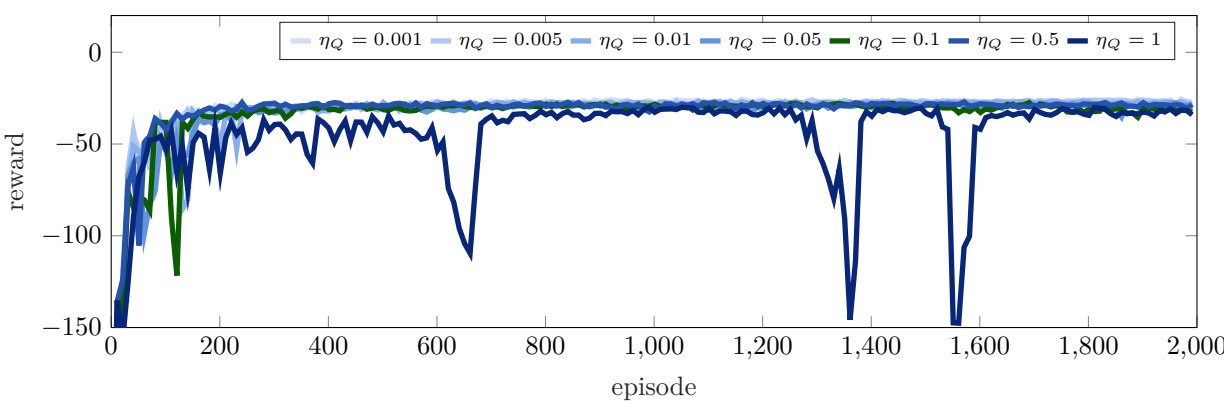

Figure 14: Learning behavior for hyper parameter ablation $\eta_Q$.

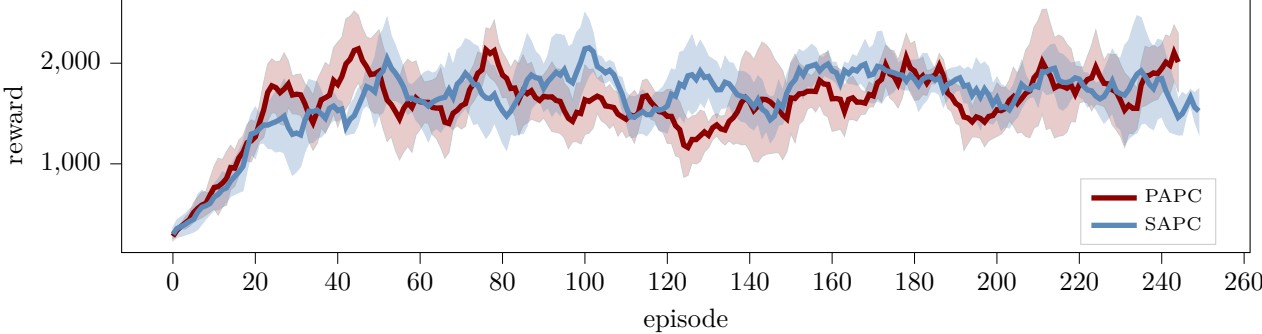

Figure 15: Full learning history for Hopper-v2.

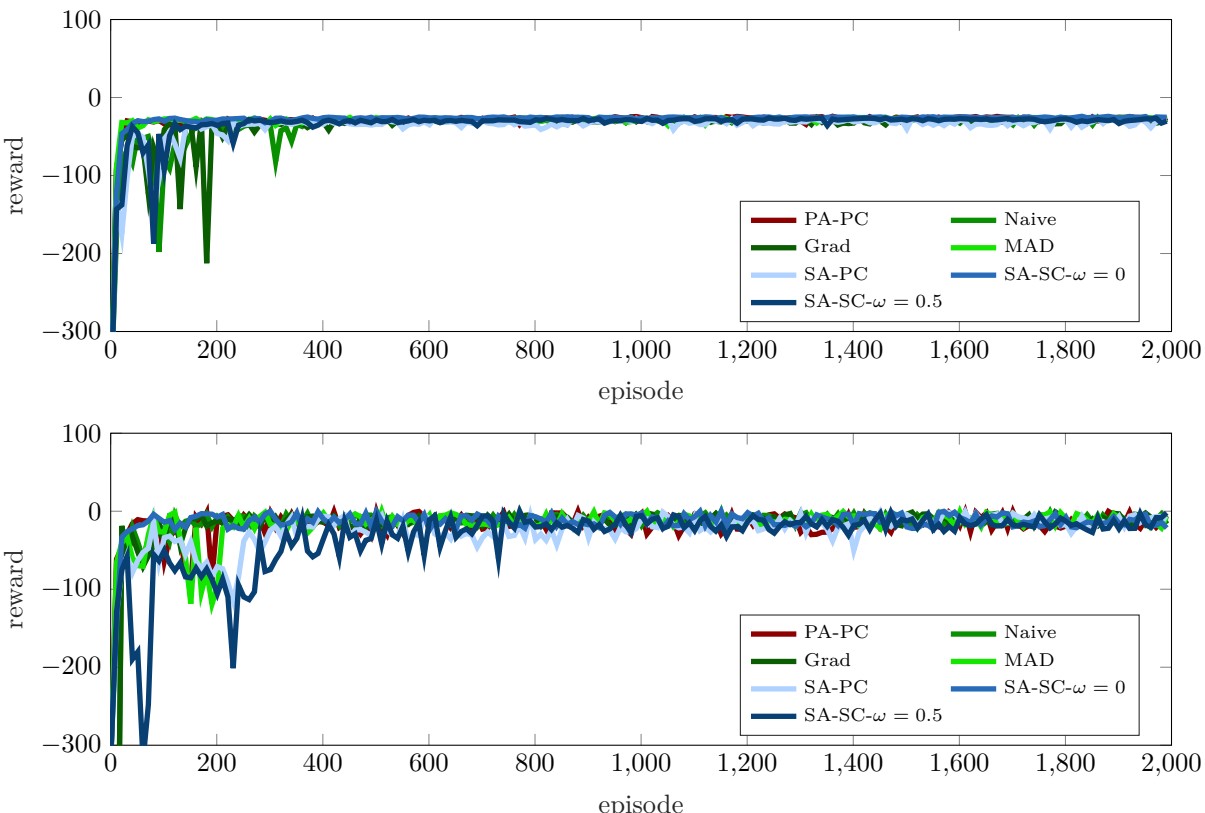

Figure 16: Full learning history for (top) 1D Quadrotor and (bottom) Inverted pendulum.

algorithms using the reward without observation noise. We find that, also in the Locomotion setting, our set-based method exhibits learning dynamics similar to those of the vanilla DDPG Lillicrap et al. (2016) implementation. Based on the hyperparameter analysis in Fig. 11 and 12, we also select $\eta_\mu = 0.1$ for the *Hopper-v2* benchmark.

