# OpenReview forum: "Training Verifiably Robust Agents Using Set-Based Reinforcement Learning"
_TMLR — Accepted by TMLR_

### Review · Reviewer_no9Q · 2026-04-28

**Summary Of Contributions:**

This paper targets addressing the vulnerability of neural network policies to input perturbations and the lack of formal robustness guarantees required for safety-critical applications. It introduces set-based reinforcement learning, a novel training paradigm which propagates entire sets of perturbed inputs through the actor and critic networks, rather than relying on individual adversarial samples. Building upon DDPG, two variants of set-based methods, SA-SC and SA-SC, were proposed and evaluated on benchmarks including 1D/2D Quadrotor, Navigation Task, and Inverted Pendulum.

**Audience:**

Yes

**Audience Explanation:**

Yes, this paper would interest JMLR readers, particularly those in robust and trustworthy ML. The zonotope propagation approach is practically relevant to enhancing the robustness of neural networks when faced with noisy input or attacks.

**Claims And Evidence:**

No

**Claims Explanation:**

The major claims of the paper include:

1.  **Claim: The proposed set-based algorithms train agents that are provably more robust and formally verifiable compared to other adversarial training methods.** This is well supported. The evidence in Figure 4 demonstrates that SA-PC and SA-SC maintain verified performance at higher perturbation radii while baseline methods like PA-PC fail entirely. The reachability analysis in Figure 5 further illustrates that SA-PC produces substantially tighter reachable sets. Furthermore,

2. **Claim: The set-based approach scales to complex benchmarks.** This is partially supported. The authors provided results on Hopper-v2; however, Hopper-v2 still represents simple dynamics and is still low-dimensional in terms of the action and state spaces. To better demonstrate the applicability, the authors should consider using more complex benchmarks like Humanoid-Bench or Deepmind Control Suite. Plus, given the runtime and computational cost of calculating the gradient set, I suspect the scalability and applicability of the set-based algorithms towards more complex tasks.

3. Finally, since the essential effect of the set-based perturbation is concentrating the output set of the policy/critic networks, I believe comparisons against well-established regularization/smoothing methods [1-2] should be included to fully demonstrate the necessity of the proposed methods.

[1] Yang, Rui, et al. "Rorl: Robust offline reinforcement learning via conservative smoothing." Advances in neural information processing systems 35 (2022): 23851-23866.

[2] Li, Qiyang, et al. "Efficient deep reinforcement learning requires regulating overfitting." arXiv preprint arXiv:2304.10466 (2023).

**Requested Changes:**

1. More comparisons agains well-established regularization and smoothing methods (please refer to my comments above).

2. Experiments on higher dimensional locomotion tasks to demonstrate the actual feasibility of the proposed method.

---

> ### Author Response · Authors · 2026-05-27
>
> Dear Reviewer no9Q,
>
> Thank you for your time and detailed comments.
> Revisions are highlighted in blue in the paper for your convenience, and the specific questions are answered below:
>
> - **Scaling to higher dimensional benchmarks:** We acknowledge the comment on the computational costs arising from our set-based computations with the computational complexity analysis in Proposition 4. Following your suggestions, we also add in our revised version Tab. 4 with runtimes per 10 training episodes for all baselines on the hopper locomotion environment.
> Furthermore, we added a dedicated limitations section, noting that we cannot provide worst case performance guarantees for Mujoco environments. Formal verification of such environments is an active field of research and currently an unsolved research problem. Consequently, the scaling bottle-neck of our method and its evaluation can therefore directly benefit from future advances in this area.
>
> - **Comparison to established smoothing methods:** We thank you for suggesting a comparison with conservative smoothing methods from offline reinforcement learning. To compare our set-based approach with RORL [1], we implement RORL for online reinforcement learning, implementing the critic ensemble that includes the out of distribution penalty using adversarial observations and the smoothed Bellman update. Additionally, RORL proposes a regularized actor update, which is similar to MAD regularization [2], and is implemented using an adversarial attack to find the perturbed observation. The additional baseline is added in Figure 5 for comparison.
>
> [1] Yang, Rui, et al. "Rorl: Robust offline reinforcement learning via conservative smoothing." Advances in neural information processing systems 35 (2022): 23851-23866.
>
> [2] Zhang, Huan, et al. "Robust deep reinforcement learning against adversarial perturbations on state observations." Advances in neural information processing systems 33 (2020): 21024-21037.
>
> Best,
> The Authors

---

### Review · Reviewer_kpx9 · 2026-05-04

**Summary Of Contributions:**

The paper introduces set-based reinforcement learning, which is a general method for producing neural network control policies that are formally verifiable under bounded input perturbations. This paper give two variants of DDPG, and empirically evaluated both methods across multiple environments.

**Audience:**

Yes

**Audience Explanation:**

Reinforcement learning is an important topic among TMLR audience. Making RL methods robust is thus relevant to the TMLR community.

**Broader Impact Concerns:**

No concerns.

**Claims And Evidence:**

Yes

**Claims Explanation:**

Most of the claims are supported by proofs or empirical evidence.

**Requested Changes:**

1. On the set based gradient defined in proposition 2, definition 3 & 4. Could you give more backgrounds on Koller's work? It is non-trivial to see how this gradient constitute an ascent direction. The vanilla policy gradient is known to converge, but it is hard to see if the set based adaptation of PG will converge.
2. The algorithm execute the centre of the action set $c_{\tilde{A}_t}$, but $\tilde{A}_t$ is stored in the buffer and used to train the Q network. Wouldn't this cause an off policy issue? How is this addressed?
3. In definition 3, the gradient seems to be compiled, i.e . a term with the gradient of the critic output set's diameter depending on the actor's generator matrix. Could you include more details of how is this computed? As computing this seems to require differentiating through both networks in a coupled way.
4. How is the runtime analysis calculated? As a reader who's not familiar with Koller's work, it is not apparent to me why does algorithm 1 enjoy this rate, especially given question 3.
5. 5 random seeds makes a rather small sample for drawing conclusion from the experiments. Considering the CI on Figure 4 is quite large. Could you provide some experiment results with more random seeds?
6. From table 3, it seems like SA-SC is trained with $\epsilon=0.1$, but on Figure 4, it was evaluated for up to $\epsilon = 0.5$ in some task. Maybe it is worth pointing this out and provide some intuitive explanation?
7. In Eq. 7, $\text{dia}(\mathcal{Z})$ should be a vector but how is $|G|$ defined? If this is a matrix norm then it is a scalar.

---

> ### Author Response · Authors · 2026-05-27
>
> Dear Reviewer kpx9,
>
> Thank you for your time and detailed comments.
> Revisions are highlighted in blue in the paper for your convenience, and the specific questions are answered below:
>
> - **More background on Koller's work:** We revised Sec. 2.5 to provide more background on Koller's work, and updated Sec. 3 to better communicate our approach following this revision.
> - **Convergence:** Based on Def. 3 and Def. 4, we can view the set-based policy gradient as gradient ascent on the following regularized objective
> $J(\phi)=\mathbb{E}_{s\sim\rho}[Q(s,\mu(s))]-\frac{\eta}{\epsilon}\mathbb{E}[\omega \log \mathrm{dia}(G_A(s))+(1-\omega)\log \mathrm{dia}(G_Q(s))]$,
> where the first term corresponds to the standard deterministic policy gradient objective and the remaining terms penalize the size of the propagated action and critic output sets. Thus, our method does not optimize a fundamentally different set-valued dynamical system, but rather augments the standard policy gradient objective with differentiable robustness regularizers derived from set propagation.
> - **Off-policy issue - $\mathcal{A}$ and $c_\mathcal{A}$:** DDPG is inherently an off-policy reinforcement learning algorithm and can therefore be trained using off-policy transitions stored in the replay buffer. In our setting, we intentionally store the full action sets in the replay buffer, as they are part of the state-action information used by the critic. This allows the critic to explicitly learn to penalize overly large Q-value sets resulting from the action sets. Consequently, the use of replayed transitions is fully consistent with the off-policy nature of DDPG.
> - **Coupled network differentiation:** Yes, this is indeed computed through (set-based) differentiation through both networks. We hope our revised Sec 2.5 clarifies this question.
> - **Runtime analysis:** We formalized the runtime analysis into a dedicated proposition with a corresponding proof, so the complexity is now derived explicitly (Prop. 4ff).
> - **More random seeds:** We thank the reviewer for the suggestion regarding additional random seeds. In our experiments, we evaluate all methods across 5 random seeds, which is standard practice in reinforcement learning. We report mean performance together with 95% confidence intervals to capture variability. Importantly, across most evaluated tasks, the confidence intervals of our method do not overlap with those of the baselines (e.g., main results in Fig. 5), indicating consistent and statistically meaningful improvements.
> - **Robustness above trained perturbation:** Good catch! We indeed observed that our approach results in agents that are surprisingly robust above the trained perturbation. We added a note about this in the evaluation section.
> - **Notation:** For a matrix $|A|$, we write $|A|$ to denote the absolute values of each entry in $A$, and for a vector $v$, $||v||_p$ denotes the $\ell_p$-norm. Thus, $|G|\mathbf{1}$ sums up all absolute values row-wise, and the result is indeed a vector. We clarified this in our notation section.
>
> Best,
> The Authors

---

### Review · Reviewer_E2Yy · 2026-05-19

**Summary Of Contributions:**

Summary:

The authors propose a novel reinforcement learning (RL) algorithm by adapting recent work on set-based learning (Koller et al., 2025) to the RL setting, based on DDPG. Their approach trains the networks with entire sets of inputs and gradients, instead of using adversarial inputs. They apply their approach to different continuous control tasks, and compare with three adversarial baseline methods.








***
Strengths:
- The studied problem is quite interesting.
- The proposed method seems reasonable, and performs favorably compared to baselines.










***
Weaknesses:
- At least personally, I found parts of the paper difficult to follow, especially Section 3.
- The authors do not clearly state or discuss the limitations of their proposed approach.









***
Questions/suggestions:
- There is a proof included for Proposition 2, but not for Proposition 1?
- Just 11 pages, so there is room up to the soft page-limit of 12 pages. Is there something in the appendix that could make sense to move to the main paper?
- _"we use neural networks with ReLU activations and two hidden layers of 64 and 32 neurons for the actor and critic networks"_: Can the proposed approach be applied to more large-scale models and settings?
- Are there other limitations of the proposed method? I think there should be a separate section/subsection in the main paper discussing this. The proposed method is presumably not perfect, under what circumstances can it break down?











***
Minor things:
- The spacing before and after Algorithm 1 looks a bit odd, very tight.
- Section 2.6: The "Althoff (2010)" reference should probably be \citep instead. Same for "Lillicrap et al. (2016)" on page 9. And same also for "Manzanas Lopez et al. (2023); Brix et al. (2023)" in Section 5.
- Mark that higher is better for the metric in Fig 4 and 6 (\uparrow)?
- Section 4: Mixed use of "quadrotor" and "quadrocopter"?
- Might want to add the legend from Fig 4 and 6 also to Fig 5?
- The caption of Table 2 can probably be improved a bit.
- Page 9, "a pure action set size penalty SA-PC (Fig. 4)", doesn't quite make sense to me.
- Section 6, "In contrast, we show that while algorithms relying on adversarial inputs only": Remove "while" here?

**Additional Comments:**

The confidence in my review is quite low, as I'm not particularly familiar with RL or adversarial/robustness methods.

**Audience:**

Yes

**Audience Explanation:**

I think the studied problem is quite interesting, and that the proposed method potentially could be useful.

**Broader Impact Concerns:**

No concerns.

**Claims And Evidence:**

No

**Claims Explanation:**

I think the paper needs to include a more explicit discussion of the limitations of the proposed approach, in order to better clarify the settings in which the method may fail or underperform. In its current form, the paper at times gives the impression that the problem has been entirely solved, which I don't think is yet justified by the presented evidence. I also think the paper would benefit from a few clarifications.

**Requested Changes:**

This is a quite interesting paper that l think could be relevant for the TMLR audience.

However, I think the current version needs a more explicit discussion of limitations, and would benefit from some clarifications and modifications, see above.

---

> ### Author Response · Authors · 2026-05-27
>
> Dear Reviewer E2Yy,
>
> Thank you for your time and detailed comments.
>
> As suggested, we included a dedicated limitations section in the main body of the paper (Sec. 6) and clarified certain paragraphs throughout.
> These revisions are highlighted in blue in the paper for your convenience, and the specific questions are answered below:
>
> - **Proof of Prop. 1:** This is a known result from the neural network literature, and we added a more explicit reference to the respective paper in the proposition title, as well as overhauled Sec. 2.5 entirely to give more background.
> - **Scalability:** We divide our answer to this question into two parts: (i) scalability of the training, (ii) scalability of the subsequent verification step.
>   i. We included an analysis on the runtime complexity (Prop. 4). We observe that our approach is limited by the growth of the number of generators to represent the set ($q$). This can be reduced by transferring the less sensitive generators to an error interval such that the new propagated set becomes: $\mathcal{Z} \oplus \mathcal{I}_\text{err}$. The full gradient set is then only computed for a fixed number of remaining  generators, and the information of the error interval can be utilized, e.g., as described in [1]. Other approaches to improve scalability are also discussed in [2, Appendix A]. This makes the robust training scale to image networks.
>   ii. Crucially, our goal is verifiable robustness improvements of the deployed agent, requiring verification algorithms of the entire system. To the best of our knowledge, there exist no verification algorithms for more complex benchmarks typically used in RL (e.g., MuJoCo). As the design of these verification algorithms is out of scope for our paper, we focused on benchmarks for which verification algorithms already exist. Nevertheless, to demonstrate that this is a limitation of the verification step rather than the training process, we apply our approach to the MuJoCo Hopper-v2 benchmark and evaluate the robustness gains with adversarial attacks (see Appendix C.3 for details).
> - **Limitation:** We included a dedicated limitation section to the paper (Sec. 6).
> - **Usage of all 12 pages:** With the suggested changes by you and the other reviewers, the paper is now at its 12-page limit.
>
> Please let us know if any further questions remain.
>
> Best,
> The Authors
>
> [1] Gowal et al. "Scalable verified training for provably robust image classification." Proceedings of the IEEE/CVF International Conference on Computer Vision. 2019.
>
> [2] Koller et al. "Set-Based Training for Neural Network Verification." Transactions on Machine Learning Research. 2025.

---

> ### Comment · Reviewer_E2Yy · 2026-06-16
>
> Thank you for the response.
>
> I think the authors have provided a solid rebuttal overall. In particular, the limitations of the proposed approach have been made much more clear in the revised paper.
>
> I will recommend accept.

---

### Decision · Action_Editor_ghPF · 2026-06-29

**Recommendation:** Accept as is

**Audience:**

Yes

**Audience Explanation:**

Trustworthy ML and RL are relevant to TMLR.

**Claims And Evidence:**

Yes

**Claims Explanation:**

After the rebuttal, all the reviewers agree that all the claims have been well supported, although there is a core weakness on the scalability and computational complexity.